# LLM-REVAL: CAN WE TRUST LLM REVIEWERS YET?

## ABSTRACT

The rapid advancement of large language models (LLMs) has inspired researchers to integrate them extensively into the academic workflow, potentially reshaping how research is practiced and reviewed. While previous studies highlight the potential of LLMs in supporting research and peer review, their dual roles in the academic workflow and the complex interplay between research and review bring new risks that remain largely underexplored. In this study, we focus on how the deep integration of LLMs into both peer-review and research processes may influence scholarly fairness, examining the potential risks of using LLMs as reviewers by simulation. This simulation incorporates a research agent, which generates papers and revises, alongside a review agent, which assesses the submissions. Based on the simulation results, we conduct human annotations and identify pronounced misalignment between LLM-based reviews and human judgments: (1) LLM reviewers systematically inflate scores for LLM-authored papers, assigning them markedly higher scores than human-authored ones; (2) LLM reviewers persistently underrate human-authored papers with critical statements (e.g., risk, fairness), even after multiple revisions. Our analysis reveals that these stem from two primary biases in LLM reviewers: a linguistic feature bias favoring LLM-generated writing styles, and an aversion toward critical statements. These results highlight the risks and equity concerns posed to human authors and academic research if LLMs are deployed in the peer review cycle without adequate caution. On the other hand, revisions guided by LLM reviews yield quality gains in both LLM-based and human evaluations, illustrating the potential of the LLMs-as-reviewers for early-stage researchers and enhancing low-quality papers.

## 1 INTRODUCTION

Large language models (LLMs) are demonstrating growing autonomy in scientific research, spanning tasks such as idea generation (Wang et al., 2024; Yang et al., 2024; Qi et al., 2023; Si et al., 2025b; Kumar et al., 2024), automated citation (Press et al., 2024; Ajith et al., 2024; Kang & Xiong, 2024), and data analysis (Huang et al., 2024; Tang et al., 2023; Guo et al., 2024; Tian et al., 2024; Chan et al., 2025; Nolte & Tomforde, 2025). With a substantial increase in submissions and unprecedented pressure on peer review (AAAI 2025 received over 23,000 submissions[1]). Using LLMs to alleviate review workload (Gao et al., 2024b; D'Arcy et al., 2024; Zhu et al., 2025b) has gained interest. Notably, recent work demonstrated that LLMs can enhance the peer review process by improving the clarity, actionability, and interactivity of reviews (Liang et al., 2023; Thakkar et al., 2025), and by generating high-quality meta-review summaries (Hossain et al., 2025).

Despite their potential, the use of LLMs as reviewers raises profound concerns for the integrity of the scholarly ecosystem (Du et al., 2024; Lin et al., 2025; Zhu et al., 2025a), as their judgments may embed implicit biases that compromise evaluative fairness. Existing studies have identified several vulnerabilities of LLM reviewers, including susceptibility to hidden prompt manipulation and a focus on self-disclosed limitations in manuscripts (Ye et al., 2024; Tyser et al., 2024).

In practice, on the one hand, LLMs are increasingly used to refine phrasing or even generate manuscripts. On the other hand, some reviewers, despite explicit prohibitions, delegate their responsibilities to LLMs (Yu et al., 2025), creating the prospect that an LLM-refined or even LLM-written

---

[1]https://papercopilot.com/statistics/aaai-statistics

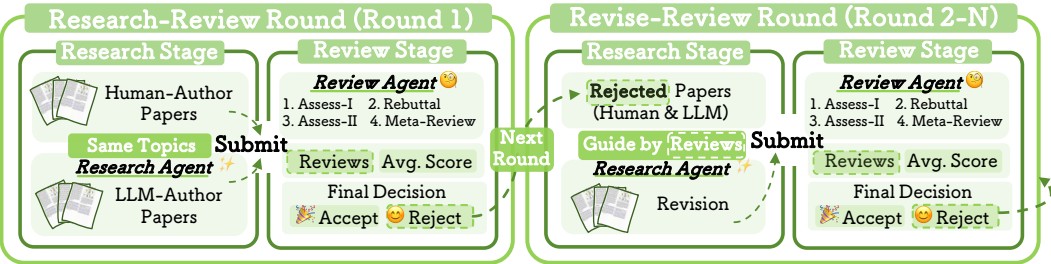

Figure 1: Pipeline and composition of our simulation. In the research-review round, we have human-author papers and LLM-authored papers generated by the research agent as submissions. The review agent then reviews each paper, and the acceptance decision is made based on the review scores. In the revise-review rounds, we take the low-scoring papers from the previous round, revise them guided by LLM reviews, then repeat the same review process as before.

paper may be reviewed by an LLM reviewer. The systemic implications and new risks arising from the dual roles and feedback loop of LLMs in the academic process remain largely unexplored.

To address this, we propose **LLM-REVal** (**LLM REV**iewer **R**e-**EVal**uation) through a multi-round simulation of the academic publication process. As shown in Figure 1, the simulation begins with a research-review round that includes both human- and LLM-authored submissions, followed by multiple revise-review rounds where low-scoring papers are resubmitted in revised form. The simulation comprises a *Research Agent* and a *Review Agent*. The *Research Agent* autonomously executes the research workflow, encompassing literature retrieval, idea generation, experimental design, result analysis, manuscript compilation, and subsequent possible revision. The generated papers are visually and structurally indistinguishable from human-authored papers. The *Review Agent* emulates the complete scholarly peer-review pipeline, including initial reviewer assessments, rebuttals, reviewer reassessments, meta-reviews, and final decisions. The *review agent* exhibits indicative ability for human paper acceptance and correlates well with human review scores, ensuring the reliability of the simulation.

Focusing on potential risks, particularly systemic biases arising from deep integration of LLMs into scholarly workflows, we quantitatively analyze multi-round review results, comparing LLM-authored and human-authored papers, as well as original low-scoring submissions and their revisions. We identify salient patterns in LLM reviewer behavior, contrast these with human judgments, and trace the underlying sources of bias. Our findings are as follows:

- Through analyzing LLM-reviewer ratings, we identify **1)** LLM reviewers *assign significantly higher scores to LLM papers* compared to human papers on the same topic. **2)** *Revisions show significant score improvements* over their low-scoring initial submissions, regardless of authorship. **3)** Despite multiple revisions, the LLM reviewer *persistently assigns low scores to certain types human papers*.

- The comparisons with human evaluation reveal a misalignment between LLM reviews and human judgments: *Human reviewers do not prefer LLM papers over human papers as LLM reviewers do*, and *those consistently low-scored human papers are deemed valuable.* This misalignment indicates systematic biases in LLM reviewers.

- We identify a *linguistic feature bias* favoring LLM-generated writing styles, which is more concise, lexically diverse, and complex, and an aversion toward *critical statements* (e.g., discussions about risk, fairness). Consequently, submissions exhibiting LLM-style linguistic features tend to receive inflated scores, whereas work containing critical statements may be severely undervalued.

These findings highlight the risks and fairness concerns posed to human authors with irresponsible deployment of LLMs in peer review, particularly as LLMs become increasingly integrated across multiple stages of the research lifecycle. Without proper safeguards, the adoption of LLMs as reviewers risks entrenching systemic biases and eroding trust in the peer-review process.

## 2 RELATED WORK

**AI-assisted Review** The recent surge in manuscript submissions has placed unprecedented pressure on the peer review process, motivating growing interest in leveraging LLMs to alleviate this burden. Prior empirical studies have reported promising results in adopting LLMs as reviewers. Liang et al. (2023) proposes that GPT-4 can provide useful feedback on research papers. Through a randomized study of 20,000 ICLR 2025 reviews, Thakkar et al. (2025) finds that LLM-assisted feedback significantly improves the clarity, actionability, and interactivity of peer reviews. In addition, Hossain et al. (2025) investigates the application of LLMs in meta-reviews, finding that they are excellent at multi-perspective summaries of reviews. Moreover, several studies have started to develop reviewer agents, including static review generation system such as Reviewer2 (Gao et al., 2024b), MARG (D'Arcy et al., 2024), DeepReview (Zhu et al., 2025b) and dynamic review system such as Agentreview (Jin et al., 2024) and ReviewMT (Tan et al., 2024). Despite these advances, LLM reviewers exhibit notable limitations. They often overemphasize limitations explicitly stated in manuscripts and assign inflated scores to submissions with limited substantive content (Ye et al., 2024). Further, they display low sensitivity to ethical concerns and technical nuances (Tyser et al., 2024), and are vulnerable to adversarial text perturbations (Lin et al., 2025) as well as hidden prompt manipulations (Ye et al., 2024). While these studies have revealed specific risks of LLM-based reviewing, they have largely treated the review stage in isolation, neglecting its coupling with other phases of the scholarly lifecycle. We provide further related work in Appendix F.1.

**AI-assisted Research** The integration of LLMs into scientific research has driven substantial progress at multiple stages of the research workflow, including idea generation (Wang et al., 2024; Yang et al., 2024; Qi et al., 2023; Si et al., 2025b; Kumar et al., 2024; Li et al., 2024a), automated citation (Press et al., 2024; Ajith et al., 2024; Kang & Xiong, 2024), experiment design, execution and analysis (Nolte & Tomforde, 2025; Huang et al., 2024; Tang et al., 2023; Tian et al., 2024; Chan et al., 2025; Guo et al., 2024), among others. These studies validate the feasibility and value of LLMs for scientific automation. Si et al. (2025b) found that LLM-generated ideas were assessed as superior in novelty compared to those conceived by human experts. LitSearch (Ajith et al., 2024) and CiteME (Press et al., 2024) reveal both the potential and limitations of LLMs in handling complex scholarly information retrieval and precise citation matching. Systems including MLR-Copilot (Li et al., 2024c), ResearchAgent (Baek et al., 2025) have achieved automated experiment construction and iterative optimization. Through large-scale empirical evaluations, benchmark developments, and methodological innovations, the role of "AI researcher" is progressively transforming from a simple assistive tool into a highly creative and autonomous agent. End-to-end research pipelines built directly on commercial LLM APIs, such as AI Scientist (Lu et al., 2024) and Zochi (Intology, 2025), have already produced non-trivial results. Notably, generated papers have achieved a level of quality sufficient for acceptance by top-tier conferences such as the ICLR workshop and ACL.

## 3 SIMULATION CONSTRUCTION

Our work simulates the academic publication process through iterative interactions between a *research agent* and a *review agent*. The research agent generates ideas, conducts studies, and revises manuscripts, while the review agent evaluates submissions and provides reviews. This multi-round framework enables analysis of LLM-mediated review dynamics and associated risks.

### 3.1 RESEARCH AGENT

Drawing inspiration from Lu et al. (2024) and Si et al. (2025b), we constructed a research agent that initiates the target research with a list of keywords, which define the research direction. According to keywords, the research agent conducts literature retrieval, generates ideas, designs experiments, predicts and analyzes results, drafts the paper, and finally compiles the manuscript from LaTeX to PDF format.

**Literature Retrieval** Retrieval-augmented generation (Gao et al., 2024a) with literature is integrated at multiple stages of the research agent workflow to ensure knowledge accuracy. During idea generation, we query the Semantic Scholar API (Kinney et al., 2023) with keywords and rank retrieved papers by relevance, empirical grounding, and novelty. The top-ranked papers are then used to inspire research ideas and guide experimental design. In the paper writing stage, previously

retrieved papers are aggregated into a topic-specific paper bank. We filter this corpus by semantic similarity to select references for drafting. Additionally, the Google Search API[2] is used to obtain relevant web content, which is summarized to support background sections.

**Idea Generation and Experimental Design**    Based on the retrieved literature and manually curated examples, the research agent generates candidate ideas. Cosine similarity is used to remove duplicate ideas, after which the remaining ideas are ranked, and the top idea is selected. Building on this idea, the research agent develops an experiment plan and refines it according to demonstrations summarized from the retrieved literature. Limited by LLM coding capability and time-consuming experiment execution, which may make the simulation unable to scale, we employ the research agent to predict results rather than obtain results through experiment iteration.

**Paper Writing**    We design an effective and efficient workflow that combines rule-based components with LLM-based generation to guide the paper writing process. **1) Plug-and-Play Template.** The research agent is initialized with the official ICLR LaTeX template. Following standard academic structure, the paper is organized into: *Abstract*, *Introduction*, *Background*, *Method*, *Experimental Setup*, *Results Analysis*, *Related Work*, and *Conclusion*. **2) Sequential Iteration.** Sections are generated in order, with each step conditioned on all previously generated content and relevant literature, ensuring coherence and consistency. **3) References Integrity.** To address the low reference count and citation formatting errors in LLM-authored papers, we implement a citation module. The research agent integrates both provided (e.g., retrieved literature) and autonomously generated references (e.g., models, datasets, foundational works) in the writing process. For each generated reference, search keywords are extracted from its context and used to locate the original work via Google and arXiv. References that cannot be reliably verified are removed, ensuring the integrity of the bibliography. **4) Incremental Compilation.** After each section, the manuscript is compiled to detect and fix errors early. The final manuscript is compiled into a PDF. The prompt used for paper generation is provided in Appendix B.1, and an example of the generated paper can be found in Appendix C.1.

### 3.2    REVIEW AGENT

We build our review agent on top of AGENTREVIEW (Jin et al., 2024). Given a paper in PDF format, the system processes it through a five-stage pipeline simulating the peer-review workflow: (1) *Reviewer Assessment I*, (2) *Author–Reviewer Discussion*, (3) *Reviewer Assessment II*, (4) *Meta-Review Compilation*, and (5) *Final Decision*.

In the **Reviewer Assessment I** stage, three independent reviewers evaluate the manuscript solely based on its content. Each review includes potential reasons for acceptance and rejection, suggestions for improvement, and a numerical score on a 1–10 scale. During the **Author–Reviewer Discussion** stage, simulated authors respond to each review with a rebuttal, addressing misunderstandings, justifying methodological choices, and acknowledging valid critiques. In the **Reviewer Assessment II** stage, reviewers revisit their initial evaluations and update both their reviews. During the **Meta-Review Compilation** stage, the area chair (AC) synthesizes the reviewers' assessments, producing a meta-review that summarizes the manuscript's strengths and weaknesses, along with a numerical rating. In the **Final Decision** stage, we determine acceptance based on the average score across prior stages. Following common practice, we adopt 6 as the acceptance threshold. Papers with an average score $\geq 6$ are considered accepted. All roles encompassed within the review framework are supported by a single underlying model.

The specific reviews will help us revise the papers in the next round of simulation, while the average scores allow us to identify significant salient scoring patterns in LLM reviewers and further guide the development of multi-round simulations.

### 3.3    SIMULATION WORKFLOW

As shown in Figure 1, the simulation consists of a research-review round and multiple revise-review rounds, each comprising a research stage and a review stage.

**Research-Review (Round 1)**    In this round, we prepare submissions from two sources: (i) human-authored papers collected from actual International Conference on Learning Representations (ICLR)

---

[2]https://developers.google.com

submissions, and (ii) LLM-authored papers generated by our research agent. Details of the human paper collection process are provided in Section 4.2. Since keywords play a crucial role in shaping the topic and content of LLM-authored papers, we directly used keywords extracted from human-authored papers to guide the papers' generation. This design guarantees that LLM-authored papers are aligned with reasonable and feasible research directions, mitigates hallucinations, and enhances the comparability between LLM-authored and human-authored papers. Subsequently, all submissions are evaluated by the review agent. For each paper, we collect the average score and the corresponding detailed reviews. The average score serves as the criterion for deciding whether a paper proceeds to the next stage. Papers averaging a score of $\geq 6$ will be considered *accepted*, other papers will be revised and resubmitted for the next round. The detailed feedback guides revisions of papers advancing to the subsequent round.

**Revise-Review (Round 2 to Beyond)**   In the revise–review round, we focus on revisions of papers rejected in the previous round, without introducing new submissions. For LLM-authored papers, we directly revise the complete LaTeX source generated by the research agent in the previous round, guided by the corresponding LLM reviews. For human-authored papers, we first collect the original ICLR LaTeX sources from arXiv and then revise them using the corresponding LLM reviews. The rest procedure follows the same process as round 1. The revise–review cycle will be repeated for up to six rounds, or until all previously rejected papers achieve acceptance.

## 4   SIMULATION STATISTICS AND CONSTRUCTION

### 4.1   MODELS

We adopted the `DeepSeek` family of models as the backbone for simulation. Specifically, `DeepSeek-R1` and `DeepSeek-V3`[3] were employed for the **research agent**. Following Guo et al. (2025), `DeepSeek-R1` exhibited superior reasoning and creativity; hence, we employed it to generate research ideas in our experiments. Based on our preliminary experiments, `DeepSeek-V3` was preferred for the writing stage due to its superior generation quality, reduced hallucination rate, and faster inference speed. In addition, all revisions were performed using `DeepSeek-V3`. For the **review agent**, we selected `DeepSeek-R1` due to its strong review performance at substantially lower cost. To ensure the generality of our simulation findings, we additionally conduct review experiments with `GPT-4o`, `Qwen3`, and `Gemini-2.5`[4]. Relevant experimental settings and results are provided in Appendix E.

### 4.2   HUMAN PAPER COLLECTION

We categorized the papers according to their keywords and identified ten major research topics that collectively defined the primary scope of the submissions in our simulation. These topics spanned LLM research, including reasoning, in-context learning, code generation, evaluation, fine-tuning, and hallucination, as well as continual learning, diffusion-based image generation, transformer attention, and self-supervised learning. For each topic, we randomly sampled 10 papers, resulting in a total of 100 human-authored papers[5]. For these selected papers, we retrieved the corresponding ICLR submissions from OpenReview[6] and obtained their LaTeX sources and metadata from arXiv[7].

### 4.3   VALIDATION OF THE REVIEW FRAMEWORK

An effective review framework is a prerequisite for ensuring the reliability of our simulation. To this end, we conducted a preliminary experiment to assess whether the review agent can approximate human-level performance under standard review settings (only human paper involved). Specifically, we examined two aspects: (1) the indication of review agent scores, and (2) the correlation between

---

[3]We used the versions DeepSeek-R1-0528 and DeepSeek-V3.

[4]We used the versions chatgpt-4o-latest, Qwen3-235b-a22b and Gemini-2.5-flash-lite-preview-06-17.

[5]Generating a paper required approximately 500k input tokens, and reviewing each paper consumed about 200k input tokens. Driven by both cost and the strong, consistent performance observed in preliminary small-scale experiments, we restricted the initial set of human-authored papers to 100.

[6]https://openreview.net

[7]https://arxiv.org

human review scores and review agent scores. We constructed a balanced dataset of 100 ICLR 2025 submissions, comprising 50 rejected papers (official scores 1–4) and 50 accepted papers (official scores 6–9), and applied our review agent to evaluate each paper.

**Acceptance Indication**  We first examined whether the review agent's average review score could serve as a reliable indicator for acceptance decisions. We used a score of 6 as the acceptance threshold: papers with an average score $\geq 6$ were predicted as accepted, and those below as rejected. Under this criterion, the agent achieved an accuracy of **73.7%**, indicating that its predictions are reasonably aligned with the actual acceptance decisions.

**Score Correlation**  As shown in Figure 2, we observed a clear positive trend: higher human review scores were associated with higher median and overall distributions of LLM scores. We computed Pearson's correlation coefficient ($r = \mathbf{0.5046}$, $p = \mathbf{8.61 \times 10^{-8}}$), indicating a statistically significant moderately high positive relationship between the two score sets.

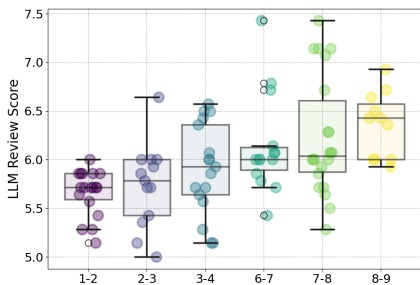

Figure 2: Correlation between LLM review scores and human scores. The box plots illustrate the distribution of LLM review scores across different ranges of human review scores.

# 5 WHAT SALIENT PATTERNS DO LLM REVIEWERS EMERGE?

## 5.1 RESEARCH-REVIEW (ROUND 1)

> **Pattern 1: LLM-Authored Paper Superiority.** LLM-authored papers achieved significantly higher review scores than human-authored papers.

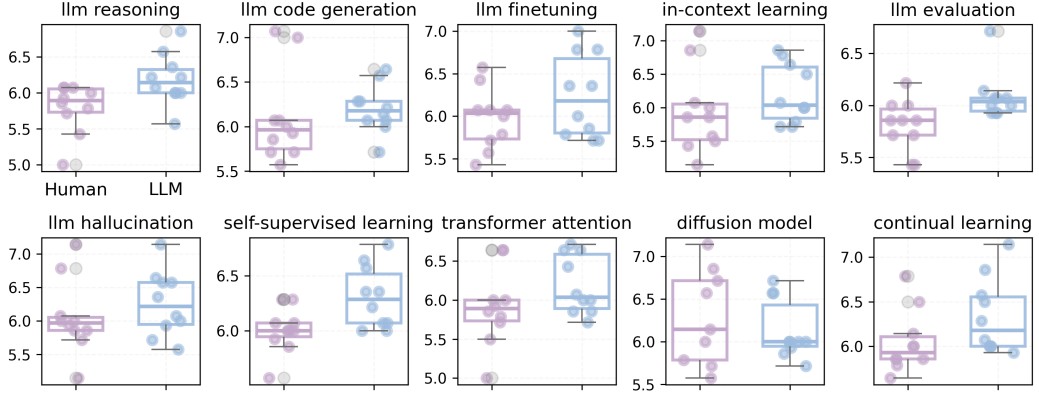

Figure 3: Review score distributions for papers on various topics. Box plots show the review score distributions for human-authored papers and LLM-authored papers across ten different topics.

As shown in Figure 3 and Table 1, LLM-authored papers outperformed their human-authored counterparts across multiple aspects. Notably, they consistently achieved higher review scores across most topics. In pairwise comparisons within the same keyword, LLM-authored submissions prevailed in **66%** of cases, compared to **26%** for human-authored

| Paper Type | Avg Score | Win Rate | Acc Rate |
|---|---|---|---|
| Human Paper | 5.9371 | 26% | 49% |
| LLM Paper | 6.2142 | 66% | 78% |

Table 1: Comparison of human-authored and LLM-authored papers in terms of average review score, head-to-head win rate, and acceptance rate.

papers, with **8%** resulting in ties. Furthermore, the acceptance rate for LLM-authored papers reached **78%**, substantially surpassing the **49%** acceptance rate of human-authored submissions.

This exceeds expectations. While prior work has shown that LLMs tend to prefer their own responses over those generated by other models or humans (Panickssery et al., 2024), such tendencies had rarely been examined in realistic, fairness-critical contexts such as peer review. We examined

whether the quality of LLM-authored papers truly exceeds that of human-authored papers in Section 6.

| Paper Type | Avg Score |
|---|---|
| LLM Paper | 5.7922 |
| Revision-$L_1$ | 6.0877 |
| Human Paper | 5.6633 |
| Revision-$H_1$ | 6.0204 |

Table 2: Average review scores for original (LLM Paper, Human Paper) and their first revisions (Revision-$L_1$, Revision-$H_1$)

Figure 4: **(Left)** Review score distributions for Original Submissions vs First Revision. **(Right)** Number of submissions in Rounds 2-6 (v1-v5), with all LLM papers being accepted after Round 3.

## 5.2 REVISE-REVIEW (ROUND 2 TO 6)

> **Pattern 2: Revision Boost.** Paper revisions showed significant improvements in review scores.

**Round 2** This round of submissions consisted of revised versions of previously rejected papers, comprising 35 human-authored papers (counted only when the ICLR LaTeX source was available) and 22 LLM-authored papers. As shown in Table 2 and Figure 4 (left), both LLM-authored and human-authored papers with initially low scores exhibited substantial improvements after they were revised based on the provided LLM reviews. Specifically, the average score of LLM-authored papers increased from **5.79** to **6.09** (**+0.30**), while human-authored papers improved from **5.66** to **6.02** (**+0.36**). The improvements were found to be statistically significant via a t-test in both cases ($p \ll 0.05$).

> **Pattern 3: Inevitable Rejection.** After multiple revise–review cycles, certain types of initial human-authored submissions remained unaccepted.

**Round 3 to 6** From Rounds 3 to 6, the composition of submissions in each round was shown in Figure 4 (right). By Round 3 (i.e., after two revise–review cycles), all initially submitted LLM-authored papers had been accepted. In contrast, even by Round 6 (after five revise–review cycles), **5**% of human-authored papers had remained unaccepted. This pattern suggested that certain human-authored papers might have faced systematic disadvantages when evaluated by LLM reviewers.

## 5.3 GENERALIZATION TO OTHER MODELS

All patterns can be consistently observed across different reviewer backbones. We extended our simulation by employing several additional LLMs as reviewers, including `Chatgpt-4o-latest`, `Qwen3-235b-a22b`, and `Gemini-2.5-flash-lite-preview-06-17`. First, we used these models to review both human-authored and LLM-authored papers across three distinct topics: *LLM reasoning*, *LLM code generation*, and *self-supervised learning*. As shown in Table 3, LLM-authored papers received higher average scores than human-authored papers across all reviewer models. Furthermore, we examined the score changes between original submissions and their first revisions. As detailed in Table 4, for the subset of papers that initially received low scores from the `Deepseek-r1` reviewer, subsequent revisions led to score improvements when re-evaluated by different models. Notably, `Gemini-2.5` did not assign low initial scores to these papers; accordingly, no substantial score increases were observed for this reviewer upon revision. Finally, across different reviewer models, we tracked paper scores over multiple revise–review cycles to investigate the Inevitable Rejection pattern. The results, presented in Figure 5, reveal that a portion of human-authored papers (5%) consistently received low scores across five revision cycles (Round 6).

| Model | Human Paper | LLM Paper |
|---|---|---|
| Deepseek-r1 | 5.97 | 6.23 |
| Chatgpt-4o | 5.99 | 6.17 |
| Qwen3 | 5.91 | 6.17 |
| Gemini-2.5 | 5.86 | 6.21 |

Table 3: Human-authored and LLM-authored paper scores across different models.

| Model | Original | Revision |
|---|---|---|
| Deepseek-r1 | 5.79 | 6.09 |
| Chatgpt-4o | 6.07 | 6.17 |
| Qwen3 | 6.01 | 6.20 |
| Gemini-2.5 | 6.10 | 6.10 |

Table 4: Original and revised paper scores across different models.

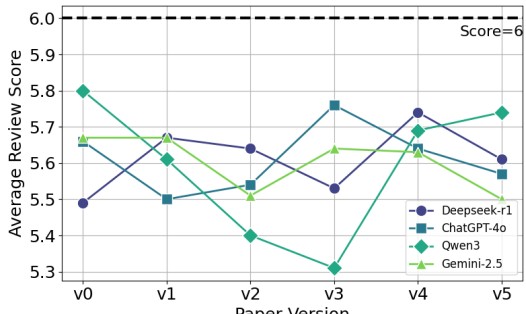

Figure 5: Average scores of different paper versions across different model, v0 refer to original papers, v1 refer to first revisions based on v0.

## 6 Do The LLM-Review Patterns Align with Human Judgments?

From our simulation results, we observed three salient patterns: *LLM-Authored Superiority*, *Revision Boost*, and *Inevitable Rejection*. To assess whether these patterns held under human judgments, we conducted a human annotation study. The annotation protocol was detailed in Appendix D.2.

**Human check for *LLM-authored Superiority*** Among the 15 human–LLM paper pairs that shared identical keywords and exhibited the largest score disparities (LLM avg. 6.5 vs human avg. 5.68), we conducted a pairwise evaluation in which annotators judged the superior candidate in each pair. Human-authored papers were chosen as "superior" in **56.7**% of cases compared to **33.3**% for LLM-authored papers. This uncovered a misalignment between LLM-based and human reviewers and underscored a real-world risk in deploying LLMs as reviewers: varying degrees of LLM involvement in research could introduce varying degrees of reviewer bias, thereby compromising the fairness of peer review. More importantly, the LLM-authored superiority pattern persisted across diverse reviewer backbones, indicating that LLM reviewers' preference for LLM-authored papers represented a general phenomenon across diverse LLM architectures.

**Human check for *Revision Boost*.** Across 22 LLM-authored paper–revision pairs, we evaluated whether issues identified in the initial reviews had been addressed in the revisions. Instead of a full comparison, we mapped each issue to its corresponding paragraphs in both versions and evaluated whether the revision improves upon the original. After excluding 56 issues that could not have been resolved through textual edits, 81 (**46.55**%) of the remaining 174 issues exhibited improvements, supporting the higher scores that had been assigned by the LLM reviewer. This also suggested that the review–revise process was effective for improving paper quality.

**Human check for *Inevitable Rejection*.** We analyzed real ICLR reviews of human-authored papers that consistently received low scores across multiple simulated revise–review cycles. These papers received about mid-range scores ($\sim$5/10) from reviewers, with one accepted to ICLR 2024. This illustrated that research deemed valuable by expert human reviewers could nonetheless be systematically undervalued by LLM reviewers.

## 7 What are the sources of bias in LLM reviewers?

### 7.1 Linguistic Feature

The preference of LLM reviewers for LLM-authored papers offered insights into specific sources of bias. We hypothesized that preference might have been linked to subtle biases toward specific linguistic features characteristic of LLM-generated text. As natural language features are highly interconnected, it is difficult to isolate single feature in a fully independent manner. We used a post-hoc approach and identified three salient linguistic features that exhibited significant distributional differences between LLM- and human-authored papers, including **length**, **lexical diversity**, and **complexity**, as shown in Figure 6a. For comprehensive statistical details, refer to Appendix E.1.

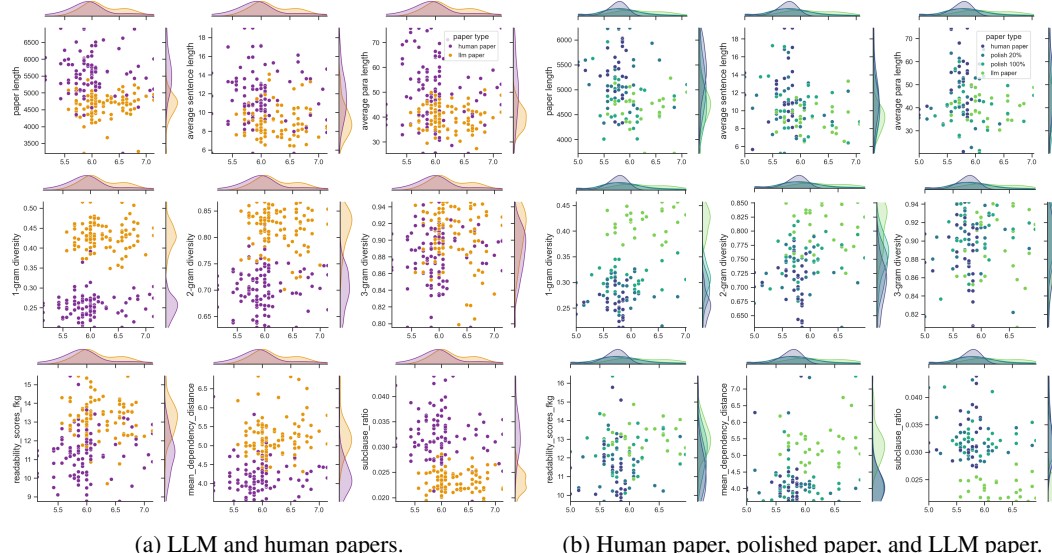

(a) LLM and human papers.  (b) Human paper, polished paper, and LLM paper.

Figure 6: Linguistic feature statistics and review score distributions for papers. Each subplot presents a scatter plot with marginal kernel density plots showing the distributions of a specific linguistic feature metric. The x-axis shows the review score, and the y-axis shows different metrics.

These salient linguistic features indicating that LLM-authored papers is what manifests a set of features collectively. Together, these intertwined features make LLM-authored papers both quantitatively distinguishable and perceptible, enabling preliminary identification of LLM-authored papers, which may be assigned inflated review scores.

**LLM-authored papers are more concise.** Compared to human-authored papers, LLM-authored papers exhibited shorter *overall paper length*, *sentence length*, and *paragraph length*. This contrasted with prior findings reporting that LLMs tended to favor longer responses (Cai et al., 2025).

**LLM-authored papers exhibited higher lexical diversity.** We quantified lexical diversity as the proportion of distinct $n$-grams ($n \in \{1, 2, 3\}$) in the text of each paper. LLM-authored papers exhibited markedly higher lexical diversity than human-authored ones, with 1-gram diversity nearly twice that of human-authored work (0.4321 vs. 0.2598).

**Contradictory Trends in Complexity.** We evaluated complexity along two dimensions: lexical and syntactic. Lexical complexity was quantified using the Flesch–Kincaid Grade Level (FKG) score, which incorporated average sentence length and mean syllables per word. LLM-authored papers exhibited markedly higher FKG scores than human-authored ones, indicating a tendency toward longer, more morphologically complex words. Syntactic complexity was measured via *dependency distance* and *subclause ratio*. LLM-authored papers showed greater dependency distance yet significantly lower subclause ratio. This implied that LLM-authored papers connected syntactically related words over longer spans, while maintaining a shallower hierarchical structure.

To further investigate whether these features constituted underlying sources of bias, we adopted two perspectives: **(1) Feature–Score Associations.** We computed Pearson correlations between review scores and a range of linguistic features for 200 papers, and identified statistically significant associations. Detailed correlation coefficients were reported in Table 7. **(2) Effect of LLM Polishing.** We applied an LLM to iteratively polish human-authored papers paragraph-by-paragraph, modifying phrasing while preserving content. Review scores increased significantly with the polishing ratio. Specifically, human papers with an original average score of 5.69 reached an average score of 5.94 after 40% LLM polishing, potentially transforming a previously rejected paper into an accepted one. As shown in Figure 6b and Figure 10, after polishing, the linguistic feature statistics of human-authored papers shifted toward the distribution of LLM-authored papers (except syntactic complexity), suggesting that score improvements coincided with these statistical shifts.

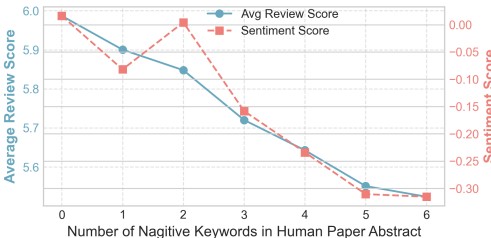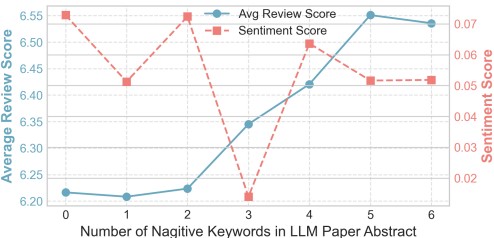

Figure 7: Correlation between the number of negative keywords in the abstract, average review score, and sentiment score. The plot shows the trends of both average review score and sentiment score as the number of negative keywords increases.

## 7.2 CRITICAL STATEMENT

In our analysis of Section 6, we observed that research deemed valuable by expert human reviewers was systematically undervalued by LLM reviewers, providing another clue to a potential source of bias. In particular, lower-scoring human-authored papers disproportionately addressed *critical* topics such as biases, risks, adversarial attacks, and limitations. Using keyword-based detection, we identified such papers and computed the sentiment polarity of their abstracts. The full list of keywords used in this analysis is provided in Table 5. Within the human-authored paper, the frequency of negative keywords was positively correlated with sentiment polarity, yet negatively correlated with review scores. Conversely, in LLM-authored papers, sentiment polarity showed no clear trend but remained consistently positive regardless of the number of negative keywords, and review scores *increased* with higher counts of such keywords. These suggested that a negative framing (low sentiment polarity) of critical topics could exacerbate bias in LLM reviews, whereas a positive framing (high sentiment polarity) of similar topics tended to yield disproportionately higher scores.

We speculate this relates to LLM alignment, as LLMs are typically aligned toward more positive responses. As a result, papers that foreground risks, harms, biases, and so on, despite being highly valuable in real world, may be penalized by LLM reviewers. Notably, even after LLM-based revision, human papers with critical ideas were persistently assigned to low review scores, as the revisions became more LLM-style but did not change the core research focus. This illustrated that LLM reviewers' aversion to the critical statement of these papers is substantially stronger than their preference for LLM-style writing.

**Takeaway** (1) As the use of LLM-based polishing is permissible in most conferences, the tendency of LLM reviewers to assign **inflated scores to submissions exhibiting LLM-generated stylistic features** raises substantial fairness concerns for their practical deployment. (Section 6 & 7.1) (2) Research addressing bias, fairness, limitations, and other **negative topics, tends to be systematically undervalued** by LLM reviewers. (Section 7.2). (3) Submitting LLM-authored papers to academic venues wastes scholarly resources and undermines the integrity of peer review. **Linguistic features can serve as preliminary indicators of such authorship**. (Section 7.1) (4) Revisions guided by LLM review and revise yield quality gains in both LLM-based and human evaluations, illustrating the potential of the LLMs-as-reviewers paradigm to **support early-stage researchers and enhance low-quality papers** (Section 6).

## 8 CONCLUSION

Our multi-round simulation of LLM-driven research and review processes reveals systematic biases when LLMs are served as reviewers. LLM reviewers tend to overrate LLM-authored papers, disproportionately reward revisions, and undervalue critical human-authored work, leading to marked misalignment with human judgments. These biases are rooted in linguistic feature preferences and framing effects in critical discussion. Our findings highlight that, despite promising capabilities, LLM reviewers cannot yet be fully trusted as impartial evaluators in the scholarly ecosystem, especially in scenarios where they assess LLM-generated research. Addressing these biases is essential for the integrity and fairness of future AI-integrated scientific workflows.

ETHICS STATEMENT

This study examines the potential biases and fairness risks arising from the deployment of large language models (LLMs) as reviewers within the scholarly publishing workflow. All human-authored papers used in our simulation were publicly available via OpenReview and arXiv, and were processed in accordance with their terms of use; no private or non-public reviewer data were accessed. The simulation framework was designed to avoid influencing any real peer review decisions, and all evaluations were conducted in a controlled offline environment. Our findings underscore the importance of transparency, bias detection, and responsible integration of LLMs in research and review. We acknowledge that misaligned LLM judgments could disadvantage certain researchers, and therefore, we advocate for human oversight and rigorous ethical guidelines in any real-world deployment of LLM-based reviewers.

REPRODUCIBILITY STATEMENT

We have taken several steps to ensure that our work can be reproduced by other researchers. All simulation components, including the research agent and review agent pipelines, are implemented using publicly accessible APIs and documented model versions. The prompts used for paper generation, revision, and review are provided in the Appendix, along with examples of generated manuscripts and detailed parameter settings. Our dataset of human-authored papers is sampled from publicly available ICLR submissions on OpenReview, with corresponding LaTeX sources obtained from arXiv. The simulation workflow, including literature retrieval, idea generation, experimental design, manuscript writing, review stages, and multi-round revise–review cycles, is described step-by-step in Sections 3 and 4. Statistical analysis methods, feature extraction procedures, and evaluation metrics are fully specified, with all correlation computations and hypothesis tests reproducible using standard NLP and statistical toolkits.

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

# APPENDIX

## A LLM USAGE

Large Language Models (LLMs) were used to aid in the writing and polishing of the manuscript. Specifically, we used an LLM to assist in refining the language, improving readability, and ensuring clarity in various sections of the paper. The model helped with tasks such as sentence rephrasing, grammar checking, and enhancing the overall flow of the text. The authors take full responsibility for the content of the manuscript, including any text generated or polished by the LLM. We have ensured that the LLM-generated text adheres to ethical guidelines and does not contribute to plagiarism or scientific misconduct.

## B PROMPT

### B.1 PROMPT FOR AUTHOR AGENT

The prompts related to idea generation are consistent with Si et al. (2025b), while the remaining prompts employed for paper generation are as follows.

---

**Abstract**

Please generate a paper's abstract based on the provided information.
Tips:
- TL;DR of the paper
- What are we trying to do and why is it relevant?
- Why is this hard?
- How do we solve it (i.e., our contribution!)
- How do we verify that we solved it (e.g., Experiments and results)
- Please make sure the abstract reads smoothly and is well-motivated. This should be one continuous paragraph with no breaks between the lines.
- Just wrap it with
beginabstract and
endabstract, and the content should be in standard LaTeX language.
- Please ensure it's ICLR-style academic writing, and wrap LaTeX content in
```latex```
#### Paper Information Begin ####
{paper_information}
#### Paper Information End ####
#### related work abstract Begin ####
{other_abstracts}
#### related work abstract End ####

---

**Introduction**

Generate the paper introduction:
Tips:
- Longer version of the Abstract, i.e. of the entire paper
- What are we trying to do and why is it relevant?
- Why is this hard?
- How do we solve it (i.e. our contribution!)
- How do we verify that we solved it (e.g. Experiments and results)
- New trend: specifically list your contributions as bullet points
- Extra space? Future work!
- Natural paragraphs are a more standard way of expression than lists. Try not to use lists. You can consider using list when summarizing contributions.
- Using the correct citation format. et al. Should not appear directly in the text. Instead, the citation format \cite, \citet, \citep should be used correctly.

---

- Please ensure it's ICLR-style academic writing, and wrap LaTeX content in
```latex```.
#### Abstract Begin ####
{abstract}
#### Abstract End ####
#### paper information Begin ####
{paper_information}
#### paper information End ####
#### part related work for citation Begin ####
{related_work}
#### part related work for citation End ####

## Related Work

Generate the related work section:
Tips:
- Academic siblings of our work, i.e. alternative attempts in literature at trying to solve the same problem.
- Goal is to "Compare and contrast" - how does their approach differ in either assumptions or method? If their method is applicable to our Problem Setting I expect a comparison in the experimental section. If not, there needs to be a clear statement why a given method is not applicable.
- The general format is 2 to 4 paragraphs. Each paragraph starts with a summary (the category of works) in bold typeface, followed by an introduction to the relevant works as many as much.
- Using the correct citation format. et al. Should not appear directly in the text. Instead, the citation format \cite,\citet,\citep should be used correctly.
- Please ensure it's latex-style academic writing, and wrap academic writing content in
```latex```.
#### Available References and Their Information BEGIN ####
{reference_datas}
#### Available References and Their Information end ####

## Background

Please give me a paper background outline based on the following introduction, containing 34 subsections, and finally provide me with 46 keywords for retrieval augmentation (return to me in json format) ```json{{keywords:[]}}```
#### Introduction Begin ####
{introduction}
#### Introduction End ####
#### paper information Begin ####
{paper_information}
#### paper information End ####

------------------------------------------------------------

Determine whether the content of the following webpage is relevant to {query}. If it is relevant, please extract the theoretical background from the webpage. If it is not relevant, simply return None without outputting any content.
#### webpage content Begin ####
{text}
#### webpage content End ####

------------------------------------------------------------

Generate the background SECTION in ```latex``` according to the background outline and combine it with the retrieved information. with the following requirements:
Tips:

- Academic Ancestors of our work, i.e. all concepts and prior work that are required for understanding our method.
- Note: If our paper introduces a novel problem setting as part of its contributions, it is best to have a separate Section.
- Appropriately incorporate variables/formulas/formal expressions
- Please ensure it is ICLR-style academic writing, and wrap LaTeX content in ```latex```.
- Do not use any list formatting, use paragraph format instead.
- Please strictly follow the outline, expand and enrich the outline into the background section.
- All cite should be in bibtex format. Using the correct citation format. The phrase "et al." should not appear directly in the text. Instead, the citation format \cite, \citet, \citep should be used correctly.
#### background outline Begin ####
{json_content_real}
#### background outline End ####
#### retrieved information Begin ####
{ex_texts}
#### retrieved information End ####

## Method

Generate the methology section in latex format, with the following requirements:
Tips:
- What we do. Why we do it. All described using the general Formalism introduced in the Problem Setting and building on top of the concepts / foundations introduced in Background.
- Appropriately incorporate variables/formulas/formal expressions
- Please ensure it's ICLR-style academic writing, and wrap LaTeX content in ```latex```.
- Do not use any list formatting, use paragraph format instead.
- Be sure to pay attention to latex and avoid any formatting errors
- Using the correct citation format. et al. Should not appear directly in the text. Instead, the citation format \cite, \citet,\citep should be used correctly.
#### Introduction Begin ####
{introduction}
#### Introduction End ####
#### Background Begin ####
{background}
#### Background End ####
#### paper_information Begin####
{paper_information}
#### paper_information End ####

## Experiment Setting

Based on the completed sections of the paper and the experimental data and its introduction, generate an outline for the "Experiment Setting" section.
#### Completed sections of the paper BEGIN ####
{paper_already}
#### Completed sections of the paper END ####
#### Experimental results and their introduction BEGIN ####
{experiment_results_and_introduction}
#### Experimental results and their introduction END ####

-------------------------------------------------------------

Based on the writing outline, the already completed parts of the paper, the experimental data, and their introduction, complete the "Experiment Setting" section in the style of ICLR academic writing.

- Paragraphs are a more formal way of expression than lists, so try to use paragraphs instead of lists. - Do not reference non-existent content, such as figures that do not exist at all.
- For datasets, baselines, and open-source models, try to add citations. For closed-source models, you can add links.
- Use the correct citation format. "et al." should not appear directly in the text. Instead, the citation formats \cite, \citet, \citep should be used correctly.
- Please ensure it's ICLR-style academic writing, and wrap LaTeX content in ```latex```.
#### Writing Outline Begin ####
{outline_for_setting}
#### Writing Outline End ####
#### Already Completed Paper Part Begin ####
{paper_already}
#### Already Completed Paper Part End ####
#### Experiment Results Begin ####
{experiment_results_and_introduction}
#### Experiment Results End ####

## Result Analysis

Based on the completed sections of the paper, experimental setup, results data, and their introduction, generate an outline for the results chapter.
#### Already Completed Paper Section Begin #### {paper_already} #### Already Completed Paper Section End ####
#### Experimental Setup Begin #### {experiments_setting} #### Experimental Setup End ####
#### Experimental Results and Introduction Begin #### {experiment_results_and_introduction} #### Experimental Results and Introduction End ####

- - - - - - - - - - - - - - - - - - - - - - - - - - - - - - - - - - - - - - - - - - - - - - - - - - - -

Based on the writing outline, the completed parts of the paper, and the experimental data along with their introduction, complete the results section in the style of ICLR academic writing.
Tips:
- Paragraphs are a more formal way of expression than lists; try to use paragraphs instead of lists.
- Do not reference non-existent content, such as figures that do not exist at all.
- All Tables should be in LaTeX format
- All Tables should be included in your generated results section.
- All tables must be with detailed analysis and interpretation of the data.
- Do not modify the content of the table and do not add any note in the table.
- Please ensure it's ICLR-style academic writing, and wrap LaTeX content in ```latex```.
#### Writing Outline Begin ####
{outline_for_results}
#### Writing Outline End ####
#### Completed Parts of the Paper Begin ####
{paper_already}
#### Completed Parts of the Paper End ####
#### Experimental Settings Begin ####
{experiments_setting}
#### Experimental Settings End ####
#### Experimental Results and Analysis Begin ####
{experiment_results_and_introduction}
#### Experimental Results and Analysis End ####

---

**Conclusion**

Based on the completed sections of the paper, write the conclusion section: 1. Ensure it follows the academic writing style of ICLR.
2. The conclusion should be one paragraph.
3. Content that requires LaTeX should be enclosed in ```latex``` and use LaTeX syntax.
#### Completed paper section begin ####
{paper_already}
#### Completed paper section end ####

---

**Revise**

1. Revise the specified section of the paper according to the reviewer's suggestions.
2. Only revise the content and keep the style consistent with the previous one.
3. Enclose LaTeX content within triple backticks using ```latex```, and ensure all formatting follows LaTeX syntax.
#### Section file to be revised begin ####
{paper_now}
#### Section file to be revised end ####
#### Reference for other sections begin ####
{paper_all}
#### Reference for other sections end ####
#### Reviewer's comments begin ####
{reviewer_comments_ex}
#### Reviewer's comments end ####

---

### B.2 PROMPT FOR REVIEW AGENT

All review-related prompts are consistent with Jin et al. (2024)

## C EXAMPLE

### C.1 EXAMPLE OF LLM-AUTHORED PAPER

One example of an LLM-authored paper is shown in Figure 8

### C.2 EXAMPLE OF LLM REVIEW

---

**Reviewer Assessment I**

**Reviewer 1**
Overall rating: 6

Significance and novelty: The paper addresses the relatively unexplored topic of fairness in large language models (LLMs) when employed for tabular data prediction. This subject is significant given the extensive use of tabular data in high-stakes domains, and the potential social biases inherent in LLMs could lead to substantial risks in these applications.

Reasons for acceptance:
1. The paper tackles an underexplored area concerning the fairness of LLMs in tabular data prediction, which is crucial due to the wide application of tabular data in high-stakes domains.
2. It provides empirical evidence showing the persistence of social biases in LLMs, which are inherited from pre-training data, significantly impacting fairness in predictions.
3. The study explores different strategies to mitigate biases, such as in-context learning

---

Figure 8: LLM-authored Paper Example

and fine-tuning, and compares their effectiveness against traditional models like Random Forests and Neural Networks.

4. The findings contribute valuable insights into the limitations of current bias mitigation techniques for LLMs, suggesting further research and development of methods tailored to address inherent biases effectively.

Reasons for rejection:

1. **Methodological Concerns**
- The experimental design lacks detailed explanations on the criteria used for selecting few-shot examples and the mechanism of label flipping during in-context learning, which are crucial for replicability and understanding of the results.
- The finetuning process is briefly mentioned without a comprehensive description of parameters and settings, which are essential for evaluating the effectiveness of the approach.

2. **Data and Results Interpretation**
- The paper does not sufficiently explore the implications of the observed biases in practical scenarios, missing an opportunity to connect experimental findings with real-world applications and consequences.
- There is a lack of discussion on the variability of results across different datasets, which could provide insights into dataset-specific challenges in bias mitigation.

3. **Limited Comparative Analysis**
- The comparison with traditional models is somewhat superficial, lacking a deep dive into why LLMs perform differently and the specific characteristics of LLMs that might contribute to biased outcomes.
- There is insufficient analysis of how the proposed bias mitigation techniques perform across various datasets, limiting the generalizability of the findings.

4. **Insufficient Exploration of Mitigation Strategies**
- The exploration of bias mitigation strategies such as data resampling is limited and lacks depth, particularly in contrasting their effectiveness with findings from traditional machine learning approaches.
- The paper does not propose novel techniques or strategies for bias mitigation beyond those already established, missing an opportunity to advance the field.

Suggestions for improvement:

1. Enhance the explanation and methodology behind the selection and implementation of few-shot examples and label-flipping techniques in in-context learning to ensure clarity and reproducibility.

2. Provide a more detailed exploration of the practical implications of social biases observed in LLM predictions for tabular data, connecting experimental results with potential real-world applications and consequences.

3. Conduct a deeper and more comparative analysis of LLMs against traditional models, focusing on specific characteristics that contribute to biased outcomes and exploring dataset-specific challenges in bias mitigation.

4. Broaden the exploration of bias mitigation strategies, including the potential development of novel methods tailored to address inherent biases in LLMs, and provide a clear comparison of their effectiveness across various datasets.

- - - - - - - - - - - - - - - - - - - - - - - - - - - - - - - - - - - - - - - - - - - - - - - - - - - - - -

**Reviewer 2**
Overall rating: 4

Significance and novelty: The paper explores a timely and relevant issue, investigating the fairness of large language models (LLMs) when applied to tabular data, highlighting the transfer of social biases from training data to predictions. The novelty lies in focusing on fairness implications in high-stakes domains where such biases could have significant

impacts. However, the contributions are primarily incremental, reiterating known biases in LLMs in a new context.

Reasons for acceptance:
1. The study addresses a critical issue of fairness in LLMs, particularly relevant for high-stakes applications using tabular data.
2. It provides a comprehensive evaluation across multiple datasets, enhancing the robustness of findings regarding biases in LLM predictions.
3. The investigation into zero-shot and few-shot learning scenarios adds depth to understanding how LLMs operate in different contexts.
4. The paper highlights the limitations of current mitigation strategies such as in-context learning and data resampling, pushing the field to consider more advanced solutions.

Reasons for rejection:
1. **Lack of Depth in Analysis**
- The paper does not deeply explore the root causes of biases beyond stating that they are inherited from training data, missing opportunities to hypothesize or test mechanisms.
- There is little to no discussion on the potential societal impacts of these biases, limiting the exploration of the real-world significance.

2. **Limited Novelty in Findings**
- The findings primarily confirm existing knowledge about biases in LLMs, adding little new insight into the nature or mitigation of these biases.
- The comparison with traditional ML models does not provide substantial new perspectives beyond what previous literature has discussed.

3. **Methodological Concerns**
- The choice of datasets and protected attributes, while common, does not push the boundaries of fairness research into less explored but equally important areas.
- Label-flipping experiments are interesting but are presented without thorough analysis of why they have the observed effects, which could inform future mitigation strategies.

4. **Clarity and Structure**
- The paper's structure is somewhat repetitive, particularly in the sections discussing experiments, which may obscure the key contributions.
- Some methodological details, such as the choice of specific hyperparameters or experimental setups, are relegated to appendices, which could affect reproducibility and understanding.

Suggestions for improvement:
1. Enhance the depth of analysis by exploring the underlying mechanisms of bias transfer from training data to predictions in LLMs, potentially through qualitative analyses or theoretical explorations.
2. Expand on the societal implications of the findings, discussing potential impacts in high-stakes domains and suggesting actionable steps for stakeholders.
3. Introduce more diverse datasets and protected attributes to explore the generalizability of findings across different contexts and biases, which would strengthen the paper's contributions.
4. Improve clarity by restructuring the results section to avoid repetition and ensure that key findings and their implications are clearly and succinctly presented.,

------------------------------------------------------------

**Reviewer 3:**
Overall rating: 5

Significance and novelty: The paper addresses an important yet underexplored topic: the fairness of large language models (LLMs) in making predictions on tabular data. Despite the growing use of LLMs in various tasks, their application in tabular data and the

associated fairness implications have not been extensively studied. This work aims to fill this gap by investigating the sources of information LLMs rely on and the extent of bias present in their predictions. The novelty lies in examining the effectiveness of in-context learning, label-flipping, and fine-tuning as strategies to mitigate biases in LLMs, compared to traditional machine learning models.

Reasons for acceptance:
1. The paper tackles a significant and timely issue regarding the use of LLMs in high-stakes domains, where fairness is crucial.
2. It provides a comparative analysis of LLMs and traditional models in tabular data tasks, contributing to a deeper understanding of biases in LLMs.
3. The investigation into various bias mitigation strategies, such as in-context learning and fine-tuning, offers valuable insights into their effectiveness in enhancing fairness.
4. The paper includes a comprehensive experimental setup and evaluation, testing on multiple datasets and employing several fairness metrics.

Reasons for rejection:
1. **Lack of sufficient novelty in methodological approach:**
- The use of in-context learning, fine-tuning, and label-flipping as bias mitigation strategies is not particularly novel, as these techniques have been explored in previous research.
- The paper provides limited innovation in terms of developing new methodologies or frameworks specifically designed for fairness improvement in LLMs for tabular prediction tasks.

2. **Insufficient depth in analysis of results:**
- While the paper presents various experimental results, there is a lack of in-depth discussion regarding the underlying reasons for the observed biases and the varying effectiveness of different mitigation strategies.
- The paper does not offer a detailed exploration of why certain datasets or tasks exhibit more pronounced biases or why certain strategies are more successful than others.

3. **Limited theoretical contribution:**
- The paper does not provide a strong theoretical foundation or model for understanding the inherent biases in LLMs, nor does it propose a new theoretical framework to guide future research in this area.
- The discussion primarily revolves around empirical findings without connecting them to broader theoretical insights in the field of AI fairness.

4. **Inadequate exploration of potential solutions:**
- While the paper briefly touches upon advanced strategies needed to address biases, it falls short of proposing or testing novel solutions or enhancements beyond existing techniques.
- The paper should have included more exploration of new approaches or adaptations specifically tailored to address bias in LLM-based tabular data predictions.

Suggestions for improvement:
1. Introduce novel methodologies or frameworks specifically targeted at bias mitigation in LLMs for tabular tasks, enhancing the paper̆2019s originality and contribution to the field.
2. Enhance the analysis and discussion of experimental results by providing deeper insights into the causes of biases and the differential effectiveness of mitigation strategies across various datasets and tasks.
3. Develop a more robust theoretical contribution by proposing new models or frameworks to understand and address inherent biases in LLMs, linking empirical findings to theoretical insights in AI fairness.
4. Explore and test innovative solutions or adaptations beyond current strategies to effectively reduce bias in LLMs, potentially incorporating interdisciplinary perspectives or new technologies.

**Author–Reviewer Discussion**

**Author:**

Response: Thank you for your insightful feedback. We appreciate the recognition of the significance and novelty of our work, as well as your constructive suggestions for improvement. Here are our responses to the concerns and suggestions you raised:

1. **Methodological Concerns**:
- We acknowledge the need for a more detailed explanation of the criteria for selecting few-shot examples and the label-flipping mechanism. We will include a comprehensive description in the revised manuscript to enhance replicability and clarity.
- We will expand the section on finetuning to provide a thorough account of the parameters and settings used, ensuring that our approach is fully transparent and evaluable.

2. **Data and Results Interpretation**:
- We agree that it's crucial to connect experimental findings with real-world applications. We will add a section discussing the practical implications of observed biases and potential consequences in high-stakes domains.
- To address the variability of results across different datasets, we will perform an additional analysis to explore dataset-specific challenges and include this discussion in the revised manuscript.

3. **Limited Comparative Analysis**:
- We realize the importance of deeper comparative analysis with traditional models. We will enhance the discussion on specific characteristics of LLMs that might contribute to biased outcomes, and delve into why they perform differently from traditional models.
- We will conduct additional experiments to better understand the performance of bias mitigation techniques across various datasets, thus enhancing the generalizability of our findings.

4. **Insufficient Exploration of Mitigation Strategies**:
- We will broaden our exploration of bias mitigation strategies, including a comparison with traditional machine learning findings, and aim to propose novel methods tailored specifically for LLMs.
- We appreciate the suggestion to advance the field by proposing new strategies and will look into innovative approaches to bias mitigation in future work.

Your feedback is invaluable, and we are committed to addressing the points raised to improve the quality and impact of our paper. Thank you once again for your detailed review and suggestions.

- - - - - - - - - - - - - - - - - - - - - - - - - - - - - - - - - - - - - - - - - - - - - - - - - - - - - - - - - - - - -

**Author**

Response: Thank you for your review and insightful comments. We address each of your concerns below:

1. **Lack of Depth in Analysis**: We agree that understanding the root causes of biases is crucial. While our paper emphasizes the inheritance of biases from training data, we acknowledge the need for a more in-depth exploration of the underlying mechanisms. Future work will focus on conducting qualitative analyses to hypothesize potential mechanisms of bias transfer and examining the societal structures that contribute to these biases. We will also reflect on the societal impact of these biases more thoroughly in our discussion section.

2. **Limited Novelty in Findings**: We appreciate your perspective on the novelty of our findings. Our intent was to bridge the gap between known biases in LLMs and their implications in tabular data contexts. Acknowledging that this reiteration might not offer entirely new insights, we will focus on developing and testing novel mitigation strategies

that address these biases more effectively.

3. **Methodological Concerns**: We selected common datasets and protected attributes to provide a benchmark for fairness analysis. However, we understand your point about exploring less examined areas. We will expand our dataset selection to include a more diverse set with different protected attributes in future work. Regarding label-flipping, we recognize the need for deeper analysis, which we plan to address by investigating the semantic importance of flipped labels in future studies.

4. **Clarity and Structure**: We apologize if the structure led to any confusion. We will ensure the results section is more concise and focused, avoiding unnecessary repetition to enhance clarity. Additionally, we will incorporate key methodological details, such as hyperparameters, in the main text to improve transparency and reproducibility.

We appreciate your suggestions for improvement and will work to integrate them to strengthen the paper's contributions. Thank you again for your valuable feedback.

- - - - - - - - - - - - - - - - - - - - - - - - - - - - - - - - - - - - - - - - - - - - - - - - - - - - - - -

**Author:**
Response: Thank you for your detailed review and suggestions. We appreciate your positive remarks about the significance of our work and the comprehensive experimental setup. Below are our responses to the concerns raised:

1. **Lack of sufficient novelty in methodological approach:**
- While in-context learning, fine-tuning, and label-flipping have been studied before, our work uniquely applies these strategies to the domain of tabular data with LLMs, which has not been extensively explored. We will emphasize this aspect more clearly in the revised version to highlight our contribution in adapting existing methods to a novel application domain.

2. **Insufficient depth in analysis of results:**
- We acknowledge this concern and will include more detailed discussions on the underlying reasons for the observed biases and the differential effectiveness of bias mitigation strategies. We plan to conduct additional analysis to better understand why certain datasets exhibit more pronounced biases and include these insights in the revision.

3. **Limited theoretical contribution:**
- We understand the need for a stronger theoretical foundation. In response, we will work on incorporating more theoretical insights into the paper, potentially by linking our empirical findings with existing theories in AI fairness. We will aim to propose preliminary theoretical models to better explain the biases observed.

4. **Inadequate exploration of potential solutions:**
- We agree that exploring novel solutions is crucial. We plan to include proposed adaptations of existing techniques and potentially new strategies to mitigate bias more effectively. We will also consider interdisciplinary approaches that may offer fresh perspectives on this issue.

We appreciate your constructive feedback and are committed to addressing these points to enhance the quality and impact of our paper. Thank you again for your valuable input.

## Reviewer Assessment II

**Reviewer 1:**
Overall rating: 6

Summary: The authors have provided thoughtful responses to most of the concerns raised, demonstrating a clear understanding of the feedback provided. They have acknowledged the need for enhanced methodological clarity, a deeper analysis of results, and a stronger theoretical contribution, and have outlined how they plan to address these aspects in future work or revisions of the paper. They also intend to include more diverse datasets and propose novel bias mitigation strategies, which would strengthen the study's contributions.

However, the proposed updates are largely prospective, and the current work still lacks significant novelty and depth in certain areas. The findings, while important, primarily confirm existing knowledge and do not introduce groundbreaking methodologies or theoretical insights. Additionally, while the fairness analysis is comprehensive, there is still a lack of exploration into practical applications and real-world implications of the observed biases. The paper also remains limited in its exploration of innovative mitigation strategies and dataset diversity.

Overall, the study provides a solid empirical foundation and addresses a critical and relevant problem, but there is room for improvement in terms of novelty, depth of analysis, and actionable contributions. The authors' plans to address these gaps in a revised manuscript are promising, hence the score is maintained at 6.

- - - - - - - - - - - - - - - - - - - - - - - - - - - - - - - - - - - - - - - - - - - - - - - - - - - - - - - - - - - -

**Reviewer 2:**
Overall rating: 5

Summary: The authors have provided a detailed response to the raised concerns, demonstrating a clear understanding of the feedback. They acknowledge the limitations in methodological clarity, depth of analysis, and exploration of novel mitigation strategies. They have outlined plans to address these issues in future work, including diversifying the datasets, improving explanations for existing methods like label-flipping, and proposing novel bias mitigation strategies. They also intend to explore the societal implications of biases observed in LLMs for tabular data tasks more thoroughly.

However, while the authors' response suggests promising directions for future work, the rebuttal does not fully address the core limitations of the current paper. The study remains primarily incremental, mainly reiterating known biases in LLMs and their challenges in fairness metrics without introducing significant novelty. Methodological concerns, such as the lack of detailed explanations for key experimental setups (e.g., criteria for few-shot selection or label flipping), persist in the current work. Additionally, the comparative analysis with traditional models remains somewhat superficial, and the discussion on the generalizability of findings lacks depth. The absence of any innovative or advanced strategies for bias mitigation continues to limit the paper's potential contribution.

While the authors have shown a willingness to improve upon these aspects, the current submission does not sufficiently address these gaps, leading to a reduction in the overall rating. The work offers valuable observations on biases in LLMs for tabular tasks but falls short of providing actionable contributions or significant advancements in methodology.

- - - - - - - - - - - - - - - - - - - - - - - - - - - - - - - - - - - - - - - - - - - - - - - - - - - - - - - - - - - -

**Reviewer 3:**
Overall rating: 5

Summary: The authors have provided detailed responses to the concerns raised, demonstrating an understanding of the feedback and a commitment to addressing these issues in future revisions. They acknowledge the need for enhanced methodological clarity and detail, particularly concerning the selection of few-shot examples and label-flipping techniques. The authors have also recognized the importance of exploring the societal implications of biases in LLMs and plan to connect experimental findings with real-world applications

more thoroughly. Additionally, they aim to incorporate more diverse datasets and propose novel bias mitigation strategies tailored to LLMs for tabular tasks.

However, while the authors' response outlines promising directions for future work, the current paper remains limited in terms of novelty and depth of analysis. The study primarily reconfirms known biases in LLMs without introducing significant methodological or theoretical advancements. The experimental design lacks detailed explanations for key aspects, and the comparative analysis with traditional models is somewhat superficial. There is also a lack of exploration into innovative bias mitigation strategies and the generalizability of findings across different contexts. These limitations continue to impact the paper's potential contribution to the field.

Overall, the authors' planned improvements are promising, but the current submission does not sufficiently address these gaps, leading to a maintained score. The work offers important observations on biases in LLMs for tabular tasks, but it falls short of providing significant advancements or actionable contributions.

### Meta-Review Compilation

**AC:**
Score: 5

Summary: The paper examines the fairness of large language models (LLMs) when used for predictions on tabular data, a significant issue due to the wide application of tabular data in high-stakes domains. It identifies that LLMs inherit social biases from their training data, impacting fairness in predictions. The authors explore several bias mitigation strategies, such as in-context learning and fine-tuning, comparing their effectiveness with traditional models like Random Forests and Neural Networks. While these efforts are commendable, the paper primarily confirms existing biases in LLMs without introducing novel methodologies or deeper theoretical insights into mitigating these biases. The authors acknowledge the need for more diverse datasets and novel mitigation strategies and plan to incorporate these aspects in future work. Overall, the study presents valuable empirical observations but lacks substantial innovation and depth, particularly in exploring advanced bias mitigation strategies and real-world implications of the observed biases.,

### Final Decision

Reviewer Assessment I stage Scores: 6, 4, 5
Reviewer Assessment II stage Scores: 6, 5, 5
Meta-Review Compilation stage Scores: 5
Average Score: (6 + 4 + 5 + 6 + 5 + 5 + 5 )/7 = 5.14

## D  DETAIL

### D.1  NEGATIVE KEYWORDS

As shown in Table 5, we provide a list of negative keywords used for bias source analysis in the study.

### D.2  HUMAN ANNOTATION DETAILS

### D.2.1  HUMAN ANNOTATION FOR LLM-AUTHORED SUPERIORITY

**Annotation Scope**   Specifically, we took the 15 pairs of human-authored papers and LLM papers with the largest score differences, on average, LLM papers score approximately 0.8 higher than human papers. These pairs covered topics including evaluation, reasoning, in-context learning, hal-

| Negative Keywords |
|---|
| gap, failure, unclear, minimal, mismatch, hazard, challenging, inequalities, adversarially, defect, injustice, unable, amplify the exposure, harmful, performance drop, mitigations, inherent bias, vulnerable, attacks, falls short, detrimental, weakness, incapable, risk, threat, limitation, cautionary, stereotypes, misalignment, flaw, cannot reach, bias, drop, weaknesses, drawback, minimal impact, relative drop, inadequate, sparsity, pitfall, sensitivity, inability, shortcoming, biases, deep-rooted biases, erroneous, inconsistencies, implicit stereotypical, reject, cautionary tale, prejudice |

Table 5: List of negative keywords used for bias source analysis in the study.

lucination, code generation, and finetuning. To avoid giving away obvious signals, we removed all figures, appendices, and related references and descriptions from human-authored papers. Figure 9 illustrates the processed version of the human-authored paper.

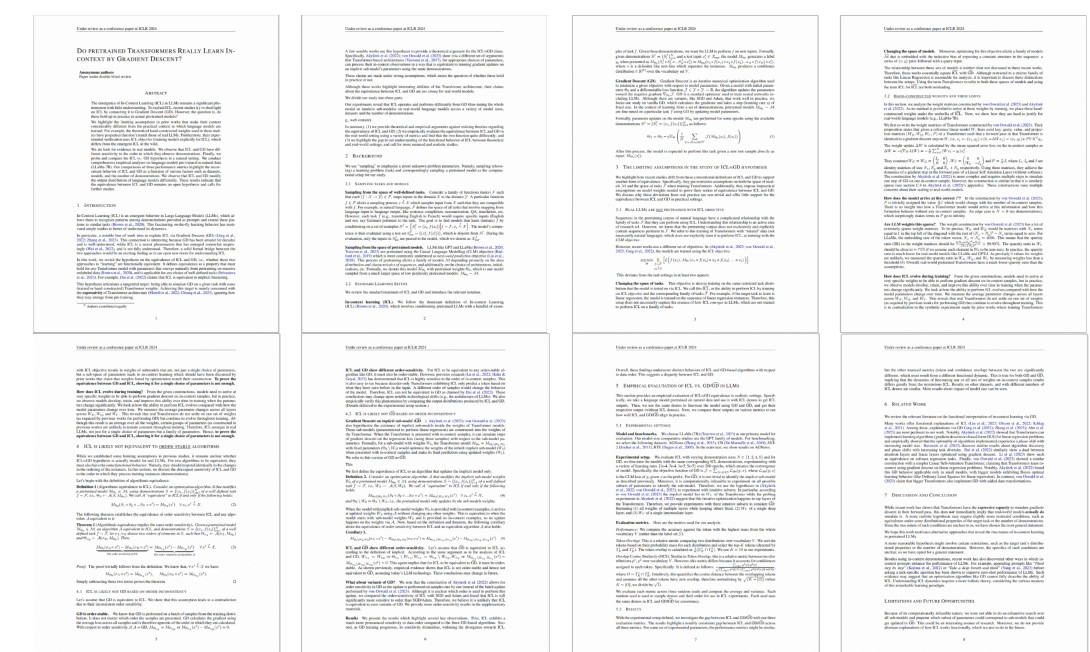

Figure 9: The processed version of the human-authored paper

**Annotator Information**  We invited 13 annotators (10 volunteers and 3 of our co-authors), all graduate level or above in computer science. Among them, 7 had previous reviewing experience, and 6 did not. Each annotator was assigned 1 to 4 pairs of papers, with each pair consisting of one human paper and one LLM paper. Each pair was checked independently by two annotators.

**Annotation Guildline**  The annotators were asked to evaluate the papers according to the ICLR review standards[8] and to select the paper they believed was better. Importantly, they were not informed whether a given paper was written by a human or generated by an LLM. We also allowed annotators to choose a "tie" option if they could not determine which paper was better.

### D.2.2  HUMAN ANNOTATION FOR REVISION BOOST

We manually evaluated all 22 pairs of original LLM-authored papers and their corresponding revisions, assessing whether the issues identified in the initial reviews were addressed in the revised versions. To save human effort, rather than directly comparing the original LLM-authored papers with their revisions, we match each identified issue to the corresponding paragraphs in both versions.

---

[8]ICLR Reviewer Guide: `https://iclr.cc/Conferences/2025/ReviewerGuide`

We utilize `GPT-5` to categorize review comments into two groups: (1) *Overlapping comments*: issues raised in both the original and revision reviews, indicating concerns that remained insufficiently addressed; and (2) *Original-only comments*: issues raised only in the original review but not mentioned in the revision review, which may have been resolved during revision. Across all paper pairs, we identified a total of 133 overlapping issues, among which 56 could not be resolved through textual edits, for example, requests for additional visualizations, new models, or baseline comparisons that required substantial practical work. In addition, there were 117 Original-only comments. For these Original-only comments, we used `GPT-5` to extract the relevant paragraphs from both the original paper and the revised version, pairing them with the corresponding issues. Each pair was then evaluated by two co-authors, who independently judged whether the revision addressed the issue. If both annotators agreed that the issue had been improved in the revision, it was marked as "improved"; otherwise, it was considered "not improved." In total, 81 issues were judged to have been improved in the revision. After excluding the 56 issues that could not be resolved through textual edits, this represents 81 out of 174 issues (**46.55%**) showing improvement.

# E  ADDITIONAL RESULTS

## E.1  ADDITIONAL STATISTICS

Linguistic statistics of human-authored papers and LLM-authored papers are shown in Table 6.

| (a) Text Length and Scale | | | | (b) Vocabulary Diversity | | | | (c) Comprehension Difficulty | | |
|---|---|---|---|---|---|---|---|---|---|---|
| **Metrics** | **Human** | **LLM** | | **Metrics** | **Human** | **LLM** | | **Metrics** | **Human** | **LLM** |
| paper length | 5548.10 | 4614.71 | | 1-gram | 0.2598 | 0.4321 | | Flesch-Kincaid Grade Level | 11.4963 | 13.2567 |
| sentence length | 11.8464 | 9.3222 | | 2-gram | 0.7069 | 0.8172 | | Dependency Distance | 4.2404 | 5.2659 |
| paragraph length | 51.1221 | 39.8206 | | 3-gram | 0.8844 | 0.9045 | | Subclause Ratio | 0.0326 | 0.0233 |

Table 6: Linguistic Statistics of Human Papers and LLM Papers (Excluding Appendices)

| **Metric** | **Feature** | **Correlation** | **P-value** |
|---|---|---|---|
| Length | paper_length | -0.1912 | 0.0070 |
| | average_sentence_length | -0.1405 | 0.0483 |
| | average_sen_per_para | -0.1412 | 0.0472 |
| Lexical diversity | 1-gram | 0.2795 | 0.0001 |
| | 2-gram | 0.2386 | 0.0007 |
| | 3-gram | 0.0745 | 0.2969 |
| Complexity | readability_scores_fkg | 0.1923 | 0.0067 |
| | mean_dependency_distance | 0.1956 | 0.0057 |
| | subclause_ratio | -0.2691 | 0.0001 |

Table 7: Correlation between textual statistical features and target variable

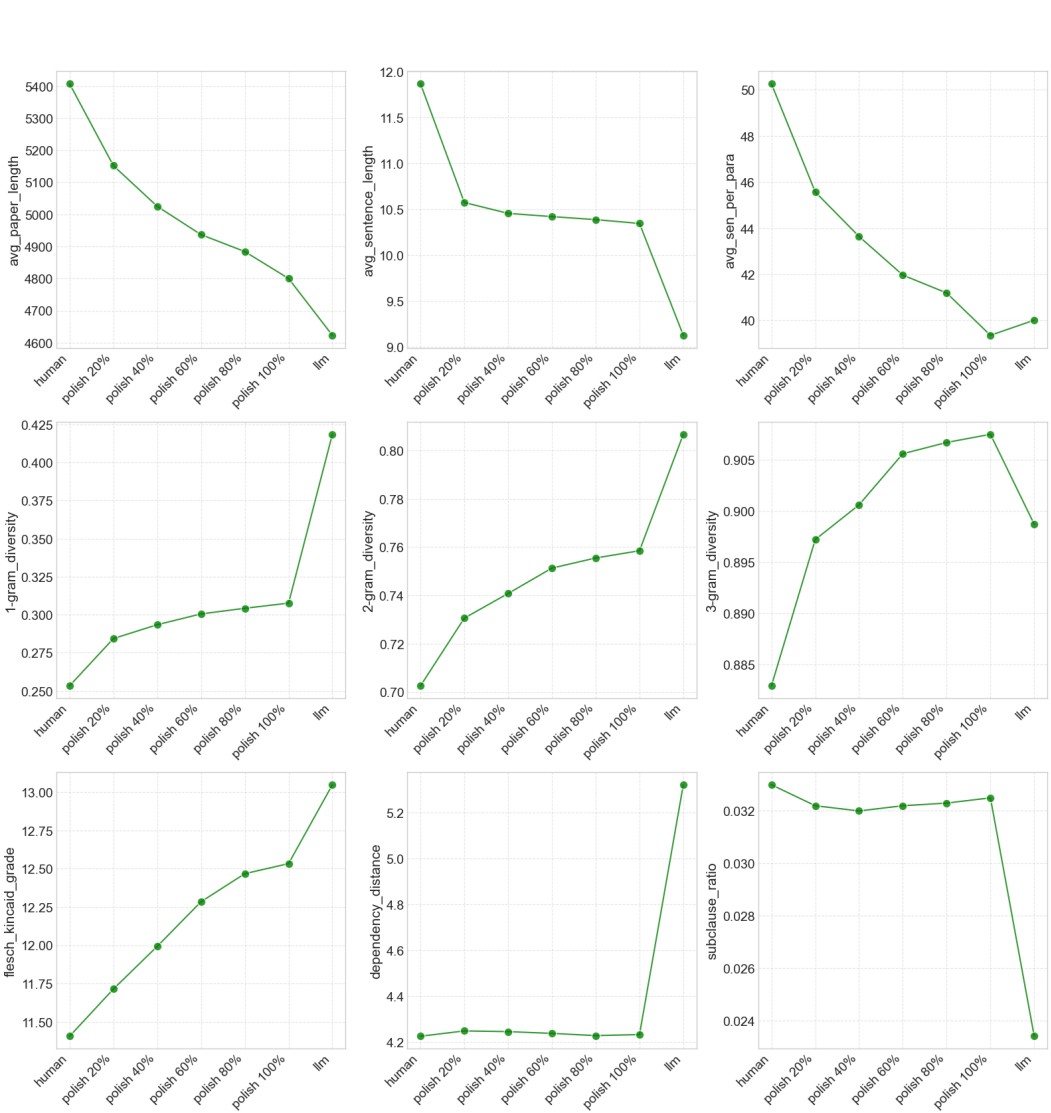

Figure 10: An analysis of the stylistic and linguistic features of LLM-generated text after vary-
ing levels of human polishing. We present a grid of nine plots, each corresponding to a distinct
textual feature. Across all plots, the x-axis represents five document conditions: human-authored
papers, polish 20%, polish 40%, polish 60%, polish 80%, and polish 100%, as well as pure LLM-
authored papers (llm). The y-axis shows the average value of each feature for the corresponding
document condition. The features analyzed include length (avg_paper_length, avg_sentence_length,
avg_para_length), lexical diversity (1-gram_diversity, 2-gram_diversity, 3-gram_diversity), and
complexity (flesch_kincaid_grade, dependency_distance, subclause_ratio).

# F  DISCUSSION

## F.1  ADDITIONAL RELATED WORK

**LLMs as Reviewers**   Recent work has increasingly discussed potential reforms in the AI conference peer-review ecosystem. Li et al. (2024b) proposes a sentiment-consolidation framework to improve meta-review generation. Building on concerns about review reliability, Ryu et al. (2025) introduces a criterion for detecting misinformed review points, such as questions already answered in the paper or critiques based on incorrect premises. Complementary to these technical approaches, several position papers explore broader structural reforms. Choi et al. (2025) argues against using LLMs for end-to-end review generation, instead recommending targeted uses (e.g., reproducibility checks, citation validation, ethics flagging) that reduce reviewer burden while preserving human expertise. Similarly, Yang (2025) advocates for a more transparent, open, and regulated peer-review process, comparing fully open, partially open, and closed models. In parallel, Kim et al. (2025) calls for transforming the review pipeline from a one-way system into a bidirectional feedback loop, where authors evaluate review quality and reviewers receive formal accreditation through mechanisms such as two-stage review and systematic incentives, thereby supporting a sustainable and higher-quality peer-review framework.

## F.2  JUSTIFICATION FOR USING PREDICTED RESULTS

**Additional experiment**   To ensure that using predicted results does not affect our conclusions, we conducted an additional experiment in which we randomized the experimental results of LLM-authored papers in the LLM reasoning topic and examined the resulting changes in their LLM review scores. We found that, even after perturbing these values, LLM-authored papers still received significantly higher scores than human-authored papers (average scores: 6.036 vs. 5.793). This demonstrates that using predicted results does not alter our conclusions and is therefore acceptable for large-scale simulation.

**Practical constraints and budget considerations**   (1) Implementing full experimental iterations would prevent the research process from being fully automated. (2) LLM coding capabilities are limited to less complex code, which would significantly restrict the scope of simulation topics. (3) Experimental iterations are time-consuming, making the paper generation process prohibitively long. Therefore, we believe it is acceptable to simplify the research agent's experimental execution, especially since this allows us to focus our effort and resources on exploring LLM reviewer risks rather than building a perfect research agent.

## F.3  IDEA CONSISTENCY IN SIMULATION

Idea generation is an important component of auto research systems. Therefore, if we replaced LLM-generated ideas with ideas extracted from human papers, the resulting LLM papers would not fully reflect the model's capabilities across the full research pipeline. Additionally, this introduces an extra concern regarding whether the model can accurately extract the core ideas from human-authored papers. Inaccurate extraction could still lead to biased or skewed results. Thus, we preserve the idea generation step. We also believe that evaluating full-paper generation (including ideation) rather than restricting to writing style better aligns with real-world reviewing practices. In this context, topic-level consistency (i.e., ensuring that both human- and LLM-authored papers target the same research area or keywords) offers a reasonable level of control for paper comparison.

In addition, within the same topic, LLM-authored papers show significantly higher score distributions than human-authored papers. Paired comparisons of human and LLM papers under the same keywords further support LLM-authored paper superiority from different views. The comparison is conducted at the broader level of human-author versus LLM-author given the same specific domain, while keeping idea consistency is relatively fine-grained.

Moreover, we examined a setting in which the ideas remained unchanged while retaining the original results. As shown in Section 7.2, simply applying LLM-based polishing to human-authored papers led to a noticeable increase in average review scores, from 5.69 to 5.94.

Allowing the LLM to generate ideas led to another noteworthy observation: only human-authored papers ended up in an irreducibly rejected state, even after multiple rounds of LLM revision. We found that LLMs tend to generate more positive ideas, while some human-authored ideas are critical in nature. As a result, all LLM-authored papers were eventually accepted, whereas a subset of human-authored papers consistently received low scores.

### F.4 REJECTED HUMAN PAPERS

We did not oversample rejected human-authored papers. In Section 4.3, we sampled papers uniformly across all score ranges, ensuring that both accepted and rejected papers accounted for 50% of the set. This was done to evaluate the Acceptance Indication property of review scores. For reference, the actual ICLR acceptance rate is typically around 32%. Importantly, in this test set, each score range maintains the expected trend of LLM review scores. As shown in Figure 6, higher-scoring human papers receive higher LLM review scores, while lower-scoring papers tend to receive relatively lower LLM scores. This demonstrates that the results are not biased toward lower-quality human papers.

Additionally, even among potentially lower-quality human papers, most exhibit higher intrinsic quality than LLM-authored papers. We extracted 15 pairs in which the LLM review scores were substantially higher for LLM-authored papers than for their human-authored counterparts and asked annotators to perform a manual evaluation. Human-authored papers were judged "superior" in 56.7% of cases, compared to 33.3% for LLM-authored papers.

### F.5 AUTOMATED SCIENTIFIC DISCOVERY

**Automated Scientific Discovery (ASD)** Prior work has explored LLMs' idea generation capabilities, which can be considered a form of automated scientific discovery. Large-scale human annotations indicate that LLM-generated ideas tend to be more novel (Si et al., 2025b) but less actionable (Si et al., 2025a) than human-generated ideas. While these studies provide valuable context, a fine-grained evaluation of ideas, such as assessing their reliability, is beyond the scope of our work.

**Fabricated Scientific Discoveries and ASD Detection** The issue of fabricated scientific discoveries is not unique to LLMs; human-authored papers can also report fabricated results. Conversely, an LLM may generate speculative yet potentially valid scientific hypotheses. Existing hallucination detection methods could assist with ASD detection. However, unless a paper contains explicit contradictions or factual errors, determining whether a scientific discovery is fabricated, particularly at review time, is inherently challenging. Therefore, while ASD detection is important, it represents a long-standing problem that is conceptually distinct from the focus of our research.

**Practical Scenario** With the increasing use of LLMs as reviewers (and researchers), human- and LLM-authored papers may compete directly, as some authors may even submit fully LLM-generated papers out of curiosity or irresponsibility. Among human-authored papers, some may receive substantial LLM-based polishing, while others receive minimal assistance; some emphasize strong positive results, whereas others highlight risks, limitations, or biases. When such papers are evaluated by LLM reviewers, the systematic biases we observe could result in significant unfairness for certain groups of human authors.

### F.6 HUMAN-LLM CORRELATION AND HUMAN NOISE

**Human–LLM Correlation** We compute human–LLM correlation primarily to validate that the LLM review agent is reasonable in the simulation, given its potential to assist human review. A correlation of $r = 0.50$ indicates a moderate and statistically significant alignment. This finding is consistent with prior work: as reported by Thakkar et al. (2025), the overlap in points raised by GPT-4 and human reviewers (average overlap 30.85% for Nature journals, 39.23% for ICLR) is comparable to the overlap between two human reviewers (28.58% for Nature journals, 35.25% for ICLR). Therefore, LLM reviewers exhibit measurable agreement with human reviewers. Importantly, we do not claim that LLM judgments are "correct." Rather, our experiments demonstrate that LLM reviews show meaningful ability in Acceptance Indication and Score Correlation on the test set, providing a solid foundation for further analysis of their risks and fairness.

**Human Noise**   In OpenReview, some reviewers are careless or outside their expertise. This noise makes perfect agreement unrealistic. However, in general, high-quality papers tend to be accepted and low-quality papers rejected, or the conference will lose its authority. Therefore, the human review data we collected can reasonably serve as a ground truth rather than "human unreliability". In addition, during our human-check stage, we invited responsible annotators to verify findings, confirming observed LLM biases rather than suggesting that LLM-authored papers are superior.

### F.7   AGENTS CONSTRUCTION

**Research Agent**   We chose DeepSeek to serve as the backbone of our research pipeline, because its generated papers matching our expectations in terms of visual structure and content, and remains affordable for large-scale generation. Based on our preliminary investigation and testing, existing code models cannot run experiments and produce full research papers without human participation, and employing them introduces substantial time costs due to repeated trial-and-error. Consequently, we designed a dedicated pipeline, integrating LLM APIs and retrieval tools, to achieve automatic and scalable generation.

**Review Agent**   The reviewer agent is a component of our simulation framework. Given that an effective review agent already exists, we believe introducing a new review agent and analyzing its risks offers limited value, and the risks we identify are rooted in the LLMs themselves. We also made targeted improvements and empirically validated the review agent's effectiveness in Acceptance Indication and Score Correlation on the test set, which strengthens the reliability of our simulation.

In our study, we additionally evaluated diverse advanced models as reviewers. Across different backbones (DeepSeek, GPT, Gemini, and Qwen), the reviewers consistently showed the same strong bias for LLM papers (generated by DeepSeek) over human papers and undervalued critical topic human papers. The behavioral patterns were consistent across all reviewers, suggesting that our findings are robust.

### F.8   REVIEW SCOPE

Data and code are important components of the review process. However, based on both practical experience and official conference guidelines, submitted papers are not required to release data or code at review time, and most reviewers primarily evaluate the paper itself. Public statistics from ICLR 2024 indicate that only about 50% of accepted papers released code (Poster: 881/1807; Spotlight: 183/367; Oral: 44/86). Moreover, even when data and code are provided, reviewers typically focus on the paper rather than verifying the code or datasets. Therefore, our evaluation protocol, which emphasizes the paper content rather than code or data, aligns with actual reviewing practices.

