# OpenReview forum: "LLM-REVal: Can We Trust LLM Reviewers Yet?"
_ICLR.cc/2026/Conference — Submitted to ICLR 2026_

### Official Review · Reviewer_TMen · 2025-10-24

**Soundness:** 3
**Presentation:** 3
**Contribution:** 4
**Rating:** 8
**Confidence:** 3

**Summary:**

The paper examines whether LLMs can function as reliable reviewers in academic peer review. To explore this, the authors develop a multi-round simulation framework called LLM-REVal, which models the interaction between a Research Agent—responsible for generating both human-like and LLM-authored papers—and a Review Agent that evaluates those submissions. The simulation reproduces the full review process, including initial assessment, rebuttals, revisions, and resubmission cycles, with a focus on fairness and bias in LLM-based reviewing.

The results reveal that LLM reviewers systematically favor the writing style characteristic of LLM-generated papers, leading to inflated scores for LLM-authored work. In contrast, some human-written papers—particularly those that discuss risks, fairness, or critical perspectives on AI—consistently receive lower scores, even after multiple revisions.

**Strengths:**

- Novel Problem Formulation: Addresses a timely and underexplored issue—what happens when LLMs act as both researchers and reviewers, creating feedback loops in scientific workflows.
- Comprehensive Simulation Framework: The paper develops a realistic multi-agent system encompassing literature search, paper creation, feedback, rebuttal, revision, and meta-review. This end-to-end simulation is technically impressive.
- Clear Empirical Evidence of Bias: The study rigorously demonstrates two forms of bias in LLM reviewers:
  - Linguistic bias toward LLM-style writing.
  - Topic/framing bias against papers emphasizing risks or fairness.

**Weaknesses:**

No significant weaknesses from my sight. Some minor comments below:

- The section title font is slightly different from other papers.
- Potential Missing Citations
  - [A Sentiment Consolidation Framework for Meta-Review Generation](https://aclanthology.org/2024.acl-long.547/)
  - [ReviewScore: Misinformed Peer Review Detection with Large Language Models](https://arxiv.org/abs/2509.21679)
  - [Position Paper: How Should We Responsibly Adopt LLMs in the Peer Review Process?](https://openreview.net/forum?id=KZ3NspcpLN)
  - [Position: The Artificial Intelligence and Machine Learning Community Should Adopt a More Transparent and Regulated Peer Review Process](https://openreview.net/forum?id=gnyqRarPzW&noteId=1Y3P0jqL5z)
  * [Position: The AI Conference Peer Review Crisis Demands Author Feedback and Reviewer Rewards](https://openreview.net/forum?id=l8QemUZaIA)

**Questions:**

NA

---

> ### Author Response · Authors · 2025-11-21
> **Response-1**
>
> We sincerely thank Reviewer TMen for your valuable and thoughtful feedback. We appreciate your recognition of the novelty of our work and your praise for our comprehensive multi-agent simulation framework  (LLMs as both researcher and reviewer). We are also grateful for your acknowledgment of the empirical evidence we provide on biases in LLM reviewers, particularly linguistic and topic/framing biases.
>
> We also thank you for your valuable suggestions. Regarding the valuable suggestions you raised:
>
> * We will review and correct the section-title font.
> * We appreciate the citations you suggested, which provide useful additional background. We will incorporate them into the revised manuscript.

---

### Official Review · Reviewer_EYY5 · 2025-10-31

**Soundness:** 2
**Presentation:** 3
**Contribution:** 2
**Rating:** 4
**Confidence:** 4

**Summary:**

This paper simulates an academic workflow to see if LLMs can be trusted as peer reviewers. The authors built a system with an LLM agent that writes papers and another that reviews them, comparing the results for both LLM-authored and human-authored papers. The study finds that LLM reviewers are significantly biased: they systematically give higher scores to other LLM-generated papers and lower scores to human papers that discuss critical topics like "risk" or "fairness".

**Strengths:**

- The paper is generally well-written and easy to follow.
- The research identifies specific biases in LLM reviewers, notably a "linguistic feature bias" favoring LLM-generated text and an aversion toward critical statements.
- The findings from the simulation are contrasted with human evaluations, which reveal a clear misalignment in judgment; human reviewers, for instance, did not share the LLM reviewers' preference for LLM-authored papers.

**Weaknesses:**

- The comparison between LLM-generated and human-authored papers is somehow not rigorous. The authors extract keywords from real human-authored papers and use these keywords to guide the LLM in generating a new paper. However, (1) the LLM may likely generate a paper with a distinct idea; (2) even with a similar idea, the LLM-generated paper uses "predicted" the results while the results in human-authored papers are real, ... That is, there are many variables that may lead to different review scores, making the comparison unfair and the corresponding conclusions less convincing.
- The research agent's process is simplified, as it "predicts" experimental results rather than actually executing experiments, which may not capture the full complexity of manuscript quality.
- The "Irreducible Rejection" finding is interesting, but it's not entirely clear why these specific human papers were persistently underrated, even after multiple revisions guided by the LLM's own feedback.

**Questions:**

n/a

---

> ### Author Response · Authors · 2025-11-21
> **Response-1**
>
> We sincerely thank Reviewer EYY5 for your thoughtful feedback and constructive comments. We are glad that you found the paper well-written. We also appreciate your recognition of the specific biases we identified in LLM reviewers. These insights are critical to our work and we are pleased that you found them meaningful.
>
> Below, we provide detailed, point-by-point responses to each of your comments.
>
> ***Q1: The comparison between LLM-generated and human-authored papers is somehow not rigorous. However, (1) the LLM may generate a paper with a distinct idea; (2) even with a similar idea, the LLM-generated paper uses "predicted" the results while the results in human-authored papers are real, ... That is, there are many variables that may lead to different review scores, making the comparison unfair and the corresponding conclusions less convincing.***
>
> Thank you for this insightful feedback. We would like to clarify the following points:
>
> * **Regarding the idea consistency:**
>   * Idea generation is an important component of auto research systems. Therefore, if we replaced LLM-generated ideas with ideas extracted from human papers, the resulting LLM papers would not fully reflect the **model’s capabilities across the full research pipeline.** Thus, we preserve the idea generation step. We also believe that **evaluating full-paper generation (including ideation) rather than restricting to writing style better aligns with real-world reviewing practices**. In this context, topic-level consistency (i.e., ensuring that both human- and LLM-authored papers target the same research area or keywords) offers a reasonable level of control for paper comparison.
>   * Within the same topic, LLM-authored papers show significantly higher score distributions than human-authored papers. Paired comparisons of human and LLM papers under the same keywords further support LLM-authored paper superiority from different views. The comparison is conducted at the broader level of human-author vs LLM-author given the same specific domain, while **keeping idea consistency is relatively fine-grained**. However, we believe that idea-level comparisons are a promising direction and may be explored further in future work.
>   * Moreover, we considered a similar setting as the reviewer suggested in our paper (​**keep ideas same and results real**​). In Section 7.1, we show that **simply applying LLM ​polishing​​ to human papers leads to a significant increase in review scores (5.69 => 5.94)​.**
>   * Allowing the LLM to generate ideas led to another noteworthy observation: only human-authored papers fell into an irreducibly rejected state, even after multiple rounds of LLM revision. We found that **LLMs tend to generate more positive ideas**, whereas some human-authored ideas are critical. As a result, all LLM-authored papers were eventually accepted, while a subset of human-authored papers consistently received low scores.
> * **Regarding the predicted results: ​**The results in LLM-authored papers do not affect our conclusions/findings. To validate this, we performed an **additional experiment** in which we **randomized** the results in LLM papers and re-evaluated them. **The review score distribution (and avg score) remained significantly higher than that of human papers (​avg LLM paper score: 6.0357, avg human paper score: 5.7929 across the topic of LLM reasoning)​**. This shows that our use of simulated results does not invalidate our conclusions; it is a reasonable choice for simulation and controlled evaluation.

---

> ### Author Response · Authors · 2025-11-21
> **Response-2**
>
> ***Q2: The research agent's process is simplified, as it "predicts" experimental results rather than actually executing experiments, which may not capture the full complexity of manuscript quality.***
>
> We acknowledge that, due to practical constraints and budget considerations, we simplified experimental execution by using predicted results. We believe this approach is acceptable in our simulation for the following reasons:
>
> * **Additional experiment: ​**To ensure that using predicted results does not affect our conclusions, we conducted an additional experiment: we **randomized** the results in LLM-authored papers and observed the changes in LLM review scores. We found that even after perturbing the numbers, LLM-authored papers still received significantly higher scores than human-authored papers (​**avg LLM paper score: 6.0357, avg human paper score: 5.7929 across the topic of LLM reasoning**​). This demonstrates that using predicted results does not alter our conclusion and is therefore acceptable for large-scale simulation.
> * **Practical constraints and budget considerations:** 1) Implementing full experimental iterations would ​**prevent the research process from being fully automated**​. 2) LLM coding capabilities are limited to less complex code, which would significantly ​**restrict the scope of simulation topics**​. 3) Experimental iterations are ​**time-consuming**​, making the paper generation process prohibitively long. Therefore, we believe it is acceptable to simplify the research agent's experimental execution, especially since this allows us to focus our effort and resources on exploring LLM reviewer risks rather than building a perfect research agent (actually, a perfect research agent would not affect our findings).
>
> ***Q3: The "Irreducible Rejection" finding is interesting, but it's not entirely clear why these specific human papers were persistently underrated, even after multiple revisions guided by the LLM's own feedback.***
>
> We appreciate your recognition of our findings. Below, we provide a detailed discussion of this finding.
>
> * "Irreducible Rejection" reflects an observable behavioral pattern. Through result analysis in Section 5.2 and the human check in Section 6, we identified a subset of papers that remained persistently underrated by LLM reviewers. Through **keyword-based detection and sentiment polarity analysis ​**​(Table 6), we observed that lower-scoring human papers tended to contain more ​**negative or cautionary language**​. We speculate this ​**relates to LLM alignment, as LLMs are typically aligned toward more positive responses**​. As a result, papers that foreground ​**risks, harms, biases​ , and so on**​, despite being **highly valuable to the scientific field, may be penalized by LLM reviewers.**
> * Even after LLM revision and polishing, human papers with critical topics were persistently assigned to low review scores. The revisions became more LLM-style but did not change the core research focus. We believe that LLM reviewers' **aversion to the critical topic framing** of these papers is ​**substantially stronger than their preference for LLM-style features**​. This explains why even after LLM polishing, the reviewers remained persistently undervalued of these papers.
>
> We sincerely appreciate your thoughtful suggestions and detailed feedback, which we will incorporate into our paper while preserving its original contributions.
>
> We hope our responses have effectively addressed your concerns. Should you have any additional questions or further feedback, we would be happy to continue the discussion. Additionally, we kindly ask you to reconsider your rating in light of our responses.
>
> Thank you again for your valuable input, and we look forward to further discussions.

---

### Official Review · Reviewer_soLf · 2025-11-01

**Soundness:** 2
**Presentation:** 3
**Contribution:** 2
**Rating:** 4
**Confidence:** 4

**Summary:**

The paper looks at the risk of LLMs as per reviewers by simulating multi round paper generation and the review process. The paper constructs a Research Agent that generates papers and a Review Agent that evaluates submissions, comparing 100 human-authored ICLR papers against 100 LLM-generated papers on identical topics across multiple review-revision cycles. The aim to then see the misalignment between LLM reviewers and human reviewers. They obseve two main biases: LLMs have linguistic feature bias favoring LLM-generated writing styles and aversion toward critical discussions. Such a study is important to discuss the course of using LLMs in peer review or not.

**Strengths:**

- The paper is very timely and helpful in the discussions of allowing AI tools for paper writing or/and paper reviewing.

- The authors validate their review agent first using real ICLR 2025 data (100 papers). The 73.7% acceptance prediction accuracy and significant correlation with human scores establish credibility and trust for downstream evaluations.

- Human in the loop validation and testing.

- While the results that LLMs prefer their own similar generations has been established in the literature previously, nine-metric linguistic analysis is thorough for review purpose.

**Weaknesses:**

- LLM papers use "predicted results" rather than actual experimental execution. This could lead to unrealistic results or bumped up values whereas human papers might have realistic results where the proposed method doesnt always outperform.

- Building upon the prev one, my biggest concern is that this study has different confounders for human paper and LLM paper, making it difficult to find causal relationship in the results.

- A bit of circular evaluation. LLMs prefer their own outputs is already known - this study essentially re-discovers it in a new context making the novelty low, especially since the reviewer agent used is also been proposed already.

**Questions:**

- With human-LLM reviewer correlation at only r=0.50 how do you establish which evaluation is "correct"? Why frame disagreement as "LLM bias" rather than "human unreliability" or "both are noisy"?

- How do you account for the confounding bias between the two LLM and human papers(paper length, overclaiming novelty)?

- If we over-sampled rejected papers, this biases results toward lower-quality human papers (as per the dataset it has 50% acceptance rate?).

---

> ### Author Response · Authors · 2025-11-21
> **Response-1**
>
> We sincerely thank Reviewer soLf for your thoughtful review and are delighted that you recognize that our work is timely in contributing to the critical discussion around the use of AI tools in academic peer review. We are also encouraged by your acknowledgment of our robust methodology, the establishment of credibility for our Review Agent, and our thorough analysis.
>
> Below, we provide detailed responses to each of your comments.
>
> ***Q1: LLM papers use "predicted results" rather than actual experimental execution. This could lead to unrealistic results or bumped up values whereas human papers might have realistic results where the proposed method doesnt always outperform.***
>
> We acknowledge that, due to practical constraints and budget considerations, we simplified experimental execution by using predicted results. We believe this approach is acceptable in our simulation for the following reasons:
>
> * **Additional experiment: ​**To ensure that using predicted results does not affect our conclusions, we conducted an additional experiment: we **randomized** the results in LLM-authored papers and observed the changes in LLM review scores. We found that even after perturbing the numbers, LLM-authored papers still received significantly higher scores than human-authored papers (​**avg LLM paper score: 6.0357, avg human paper score: 5.7929 across the topic of LLM reasoning**​). This demonstrates that using predicted results does not alter our conclusion and is therefore acceptable for large-scale simulation.
> * **Practical constraints and budget considerations:** 1) Implementing full experimental iterations would ​**prevent the research process from being fully automated**​. 2) LLM coding capabilities are limited to less complex code, which would significantly ​**restrict the scope of simulation topics**​. 3) Experimental iterations are ​**time-consuming**​, making the paper generation process prohibitively long. Therefore, we believe it is acceptable to simplify the research agent's experimental execution, especially since this allows us to focus our effort and resources on exploring LLM reviewer risks rather than building a perfect research agent (actually, a perfect research agent would not affect our findings).
>
> ***Q2&Q5: Building upon the prev one, my biggest concern is that this study has different confounders for human paper and LLM paper, making it difficult to find causal relationship in the results. How do you account for the confounding bias between the two LLM and human papers(paper length, overclaiming novelty)? ​***
>
> Thank you for your feedback on the different confounders; it gives us an opportunity to further clarify.
>
> * Admittedly, confounders objectively exist, as features of natural language are highly interconnected. However, this does not mean we cannot effectively analyze potential biases and derive meaningful relationships. Using a post-hoc approach, we found significant differences between LLM-authored and human-authored papers across ​**common linguistic features (e.g., length, lexical diversity, and syntactic complexity)​,** and these features vary with the proportion of LLM polishment in human papers, indicating that​ **LLM generation is what manifests this set of features collectively (also with other features)​**. Thus, these intertwined features **enable the LLM-author to be quantitatively distinguished and perceptible**​, helping to preliminarily identify LLM-authored papers or papers overly polished by LLMs, which may receive inflated scores. At the semantic level, the LLM's dispreference for critical topic papers was also clearly quantified.
> * In practice, it's difficult to imagine that humans prompt LLMs to "shorten the sentences," "increase lexical diversity," or "increase syntactic complexity," to improve their paper. A far more common scenario is "Please polish this paper/paragraph." The resulting polished text typically becomes shorter, more lexically diverse, and more syntactically complex. **Therefore, we argue that if these features are inherently coupled in reality, it is unnecessary to establish a one-to-one correspondence between each individual feature and the review score.**

---

> ### Author Response · Authors · 2025-11-21
> **Response-2**
>
> ***Q3: A bit of circular evaluation. LLMs prefer their own outputs is already known - this study essentially re-discovers it in a new context making the novelty low, especially since the reviewer agent used is also been proposed already.***
>
> * Our work is **not a simple re-discovery**​.
>   * **Prior findings [1,2,3,4]** on LLMs preferring their own outputs primarily focus on models evaluating content generated by themselves. In contrast, our experiments demonstrate that **different backbones (DeepSeek, GPT, Gemini, Qwen) consistently show a significant preference** for LLM-authored over human-authored papers (with LLM papers generated by DeepSeek). The behavioral patterns were consistent across all reviewers, suggesting that our findings are ​**robust and model-agnostic**​.
>   * The context addressed in our study is not only ​**new**​, but also ​**critically important**​. Historically, there have been very few settings in which LLM-generated and human-written content directly compete (especially in a way that threatens fairness). However, the situation has now changed. LLMs are becoming increasingly used (or misused) within the academic publication process. **In the current ICLR cycle, 21% of the review texts exhibited clear AI-generated characteristics [(source of information)](https://x.com/gneubig/status/1989681438577336401). An AI-generated submission received two reviewer scores of 8, suspected to be from LLM reviewers [(source of information)](https://x.com/micahgoldblum/status/1989088547777966512). This is profoundly unfair to human authors.** No prior work has systematically analyzed this problem or revealed its risks. Considering the emergent AI-review & writing issue, our study is ​**timely and necessary**​.
>   * We uncover ​**unexpected behavioral patterns​** of LLM reviewers. e.g., LLM reviewers displayed a **strong preference** for LLM-authored papers, despite research traditionally being ​**a domain of absolute human dominance**​. ​**Higher proportions of LLM polishment correlate with higher review scores**​, and LLM reviewers exhibit a ​**dispreference for critical-topic papers**​. Such behaviors, if manifested in real peer review, may have**​ long-term negative impacts on the fairness and integrity** of the review process.
> * **Building the reviewer agent on existing work does not undermine our contribution.**
>   * The reviewer agent is a component of our simulation framework. Given that an effective review agent already exists, we believe introducing a new review agent and analyzing its risks offers limited value, and the risks we identify are rooted in the LLMs themselves. We also made targeted improvements and empirically validated the review agent's effectiveness in **Acceptance Indication** and **Score Correlation** on the test set​, which **strengthens the reliability of our simulation**.
>   * Our efforts **go well beyond introducing a reviewer agent**: constructing the entire research pipeline (as we find that existing open-source research agents do not meet our quality or scalability requirements), designing a full simulation workflow, conducting comprehensive analyses of the simulation results, performing manual checks, tracing the origins of biases, and so on.
>
> [1] Large Language Models are not Fair Evaluators
>
> [2] Self‑Preference Bias in LLM‑as‑a‑Judge
>
> [3] Benchmarking Cognitive Biases in Large Language Models as Evaluators
>
> [4] Do LLM Evaluators Prefer Themselves for a Reason?

---

> ### Author Response · Authors · 2025-11-21
> **Response-3**
>
> ***Q4: With human-LLM reviewer correlation at only r=0.50 how do you establish which evaluation is "correct"? Why frame disagreement as "LLM bias" rather than "human unreliability" or "both are noisy"?***
>
> * **On the correlation score and "correct":** We compute the correlation primarily to verify that the review agent is acceptable in the simulation because it seems to have the potential to assist human review. A human–LLM correlation of r = 0.50 represents a moderate correlation and is statistically significant. Moreover, human–LLM review correlation has also been supported by prior study. As stated in [1], *"The overlap in the points raised by GPT-4 and by human reviewers (average overlap 30.85% for Nature journals, 39.23% for ICLR) is comparable to the overlap between two human reviewers (average overlap 28.58% for Nature journals, 35.25% for ICLR)."* Therefore, **correlation between LLM reviewers and human reviewers objectively exists within a certain range.** Moreover, we do not claim that the LLM reviewer's judgment is ​"**correct**".​ Rather, our experiments show that **​LLM reviews exhibit ​ability in ​​Acceptance Indication and ​Score Correlation on the test set, which provides ​a meaningful basis to further study their risks and fairness.**
>
> * **On human unreliability/noise:** In OpenReview, some reviewers are careless or outside their expertise. **This noise makes perfect agreement unrealistic.** However, in general, high-quality papers tend to be accepted and low-quality papers rejected, or the conference will lose its authority. Therefore, the human review data we collected can reasonably serve as a ground truth rather than "human unreliability". In addition, during our human-check stage, we​**​ invited responsible annotators to verify findings**​,**​ confirming observed LLM biases** rather than suggesting that LLM-authored papers are superior.
>
> [1] Can large language models provide useful feedback on research papers? A large-scale empirical analysis.
>
> ***Q6: If we over-sampled rejected papers, this biases results toward lower-quality human papers (as per the dataset it has 50% acceptance rate?).***
>
> * Actually, we did **not oversample rejected papers**​. In Section 4.3, we sampled papers uniformly across all score ranges and ensured that accepted and rejected papers each account for 50% of the set. This was done to evaluate the **Acceptance Indication** property of review scores. For reference, the actual ICLR acceptance rate is typically around 32%.  Importantly, in this test set, each score range preserves the expected LLM review score trend: as shown in Figure 2, higher-scoring human papers receive higher LLM review scores, and lower-scoring papers receive relatively lower LLM scores. This indicates that the results are **not biased toward lower-quality human papers**​.
> * Additionally, **even the possible lower-quality human papers, most of their quality is higher than LLM-authored papers. ​**We extracted 15 pairs where the LLM review scores were substantially higher for LLM-authored papers than for human-authored papers, and invited annotators to perform a manual check. Human-authored papers were judged "superior" in 56.7% of cases, compared to 33.3% for LLM-authored papers.
>
> We sincerely appreciate your thoughtful suggestions and detailed feedback, which we will incorporate into our paper while preserving its original contributions.
>
> We hope our responses have effectively addressed your concerns. Should you have any additional questions or further feedback, we would be happy to continue the discussion. Additionally, we kindly ask you to reconsider your rating in light of our responses.
>
> Thank you again for your valuable input, and we look forward to further discussions.

---

### Official Review · Reviewer_9d5H · 2025-11-07

**Soundness:** 2
**Presentation:** 2
**Contribution:** 2
**Rating:** 2
**Confidence:** 4

**Summary:**

This work studies how using LLMs for both research and review influences AI reviewing.
An AI research (agent) and an AI reviewer (agent) generate papers and review them in a loop.
The work finds that LLM reviews inflate the scores of LLM-generated papers
and penalize human-written papers that are self-critical.

**Strengths:**

1. A key strength of this work is that it studies an AI researcher (agent) together with an AI reviewer (agent).
A builder-reviewer loop is often used in the emerging field of automated scientific discovery (ASD).

2. The work finds that LLMs tend to favor LLM-written papers over human-written papers, reward revisions, and penalize human self-criticism in human-written papers.

**Weaknesses:**

1. The key issue with this work is that the review process only includes papers and is missing a review of their data and code.
Without the AI reviewing the entire submission: data, code, and paper, the AI reviewer cannot distinguish between real and fabricated papers, which may hallucinate experiments, results, etc.
Without reviewing the entire submission, it is unclear if improvements are hallucinated research or real research.

2. The architecture figure 1 (on page 2) is AI-generated with gross spelling errors, “Guild by reviews”, “Reversion”.

3. The "research agent" may be simplified by using an agent such as Claude Code (with a flat fee without incurring any API token costs).
The "review agent" could be improved by using a strong model such as GPT-5 Pro.

4. This perspective on LLM reviewing of both human and AI researchers is missing the issue of detecting problems with automated scientific discovery (ASD) systems.
See for example:
@article{jiang2025badscientist,
  title={BadScientist: Can a Research Agent Write Convincing but Unsound Papers that Fool LLM Reviewers?},
  author={Jiang, Fengqing and Feng, Yichen and Li, Yuetai and Niu, Luyao and Alomair, Basel and Poovendran, Radha},
  journal={arXiv preprint arXiv:2510.18003},
  year={2025}
}

**Questions:**

Can the work be used to evaluate issues with automated scientific discovery (ASD) systems instead of just LLM reviewing?

---

> ### Author Response · Authors · 2025-11-21
> **Response-1**
>
> We sincerely thank Reviewer 9d5H for your review and are delighted that you recognize the fundamental strength of our work and acknowledge our key empirical findings.
>
> Below, we provide detailed responses to each of your comments.
>
> ***Q1: The key issue with this work is that the review process only includes papers and is missing a review of their data and code. Without the AI reviewing the entire submission: data, code, and paper, the AI reviewer cannot distinguish between real and fabricated papers, which may hallucinate experiments, results, etc. Without reviewing the entire submission, it is unclear if improvements are hallucinated research or real research.***
>
> Thank you for the feedback!
>
> * **We agree that data and code are important components of the review process. ​**However, ​based on both practical experience and the conference's official requirements, submitted papers are not obligated to release data or code at review time. Most reviewers primarily evaluate the paper itself. According to public statistics from ICLR 2024,​**​ only about 50% of accepted papers released code ​**​(Poster: 881/1807; Spotlight: 183/367; Oral: 44/86). Moreover, even when data and code are provided, reviewers typically focus on the paper rather than verifying the code or datasets. Therefore, we believe that our evaluation protocol, focusing on paper content rather than code/data, is aligned with actual reviewing practices. Of course, incorporating data and code is a valuable direction and worth exploring in future work.
> * **On distinguishing between real and fabricated papers: ​**We agree that **distinguishing** is an important problem, but it is orthogonal to our focus. Our work examines the **fairness and risks of ​LLM​ review, not fabricated-paper detection**​. Our study finds that LLM-authored papers with substantial flaws can still receive relatively high scores from LLM reviewers, which poses a significant unfairness for human authors. Moreover, we do briefly discuss the possibility of linguistic feature-based fabricated paper detection in our paper.
> * **On whether predicted results affect our conclusions: ​**The results in LLM-authored papers do not materially influence review scores to a degree that could overturn our findings. To validate this, we performed an additional experiment in which we **randomized** the results in LLM papers and re-evaluated them. **The review score distribution remained significantly higher than that of human papers (​avg LLM paper score: 6.0357, avg human paper score: 5.7929 across the topic of LLM reasoning**​). This shows that our use of simulated results does not invalidate our conclusions; it is a reasonable choice for the purposes of simulation and controlled evaluation.
>
>
>
> ***Q2: The architecture figure 1 (on page 2) is AI-generated with gross spelling errors, "Guild by reviews", "Reversion".***
>
> We apologize for the typos in Figure 1. We have corrected them in the revised version ("guild" => "guide", "reversion" => "revision"). Additionally, Figure 1 was carefully designed and created manually in PowerPoint and then exported to PDF format; no AI tools were used at any stage. We can provide the original PPT source files if needed.
>
> We hope these minor spelling errors do not affect your overall assessment of the paper.

---

> ### Author Response · Authors · 2025-11-21
> **Response-2**
>
> ***Q3: The "research agent" may be simplified by using an agent such as Claude Code (with a flat fee​ without incurring any API​ token costs). The "review agent" could be improved by using a strong model such as GPT-5 ​Pro.***
>
> We appreciate your suggestion. Below, we provide clarifications regarding the soundness of the research agent and the rationality of the review agent.
>
> * **Regarding the research agent:**
>   * We chose **DeepSeek** to serve as the backbone of our research pipeline, because its generated papers matching our expectations in terms of visual structure and content, and remains affordable for large-scale generation.
>   * Based on our preliminary investigation and testing, **existing code models cannot run experiments and produce full research papers without human participation**​, and employing them introduces substantial time costs due to repeated trial-and-error. Consequently, we designed a dedicated pipeline, integrating LLM APIs and retrieval tools, to achieve reliable automation and scalable generation.
> * **Regarding the review agent:**
>   * In our study, we additionally evaluated diverse advanced models as reviewers. Across different backbones (DeepSeek, GPT, Gemini, and Qwen), the reviewers consistently showed the same strong bias for LLM papers (generated by DeepSeek) over human papers and undervalued critical topic human papers. The behavioral patterns were consistent across all reviewers, suggesting that our findings are ​**robust and model-agnostic**​.
>
> ***Q4: This perspective on LLM reviewing of both human and AI researchers is missing the issue of detecting problems with automated scientific discovery (ASD) systems. & Can the work be used to evaluate issues with automated scientific discovery (ASD) systems instead of just LLM reviewing?***
>
> We appreciate your raising the topic of detecting problems with automated scientific discovery (ASD) systems. We will clarify our perspective on ASD, fabricated scientific discoveries, and ASD detection, and further emphasize the focus of our work.
>
> * **On ASD: ​**Prior work has examined LLMs' idea generation (which can be seen as automated scientific discovery) capability.  Large-scale human annotations show that LLM-generated ideas tend to be more novel [1] but less actionable [2] than human ideas. These studies provide useful background, but a fine-grained evaluation of ideas (e.g., assessing their reliability) is not the focus of our work.
> * **On fabricated scientific discoveries and ASD detection: ​**The issue of fabricated scientific discoveries is not unique to LLMs. Human-authored papers can also report fabricated discoveries and results. Conversely, an LLM could generate speculative but may valid scientific hypotheses. We believe that existing hallucination detection methods may help with ASD detection. However, unless a paper contains explicit contradictions or factual errors, it is difficult to determine whether a scientific discovery is fabricated (especially at review time). Thus, ASD**​ ​**detection is a long-standing challenge; it is valuable but fundamentally separate from our research direction.
> * **Our focus: Risk and fairness in LLM reviewing. ​**With the increasing use of LLMs as reviewers (and researchers),  human and LLM-authored papers may compete directly (as some authors may submit fully LLM-generated papers out of curiosity or irresponsibility); some human-authored papers may receive substantial LLM polishment, while others receive minimal LLM assistance; some papers emphasize strong positive results, while others highlight risks, limitations, or biases. If these papers happen to be reviewed by LLM reviewers, the systematic biases we observe could create **significant unfairness** for certain groups of human authors.
>
> [1] Can LLMs Generate Novel Research Ideas? A Large-Scale Human Study with 100+ NLP Researchers
>
> [2] The Ideation-Execution Gap: Execution Outcomes of LLM-Generated versus Human Research Ideas
>
>
>
> We sincerely appreciate your thoughtful suggestions and detailed feedback, which we will incorporate into our paper while preserving its original contributions.
>
> We hope our responses have effectively addressed your concerns. Should you have any additional questions or further feedback, we would be happy to continue the discussion. Additionally, we kindly ask you to reconsider your rating in light of our responses.
>
> Thank you again for your valuable input, and we look forward to further discussions.

---

### Author Response · Authors · 2025-12-04
**General Response**

Dear (Senior) Area Chair,

We are deeply regretful about the recent information leakage within the community. To support a fair and thorough evaluation, we would like to summarize the reviews and our rebuttal. For clarity and simplicity, we will refer to Reviewers 9d5H, soLf, EYY5, and TMen as R1, R2, R3, and R4, respectively, in the following response. We sincerely thank all reviewers for their thoughtful and constructive feedback.

In particular, we are encouraged by the acknowledgment of our novel problem formulation addressing an underexplored issue (R4) and the timely contribution of our work to the ongoing discussions around the use of AI tools in academic review (R3, R4). We also value recognition of our comprehensive simulation framework, including a realistic multi-agent system (R4), and our empirical validation, which was key to demonstrating the credibility of our simulation (R2, R3). Additionally, our empirical analysis of LLM reviewers' biases has been highlighted as both important and impactful (R2, R3, R4). Finally, we greatly appreciate the reviewers' positive remarks about the clarity and structure of our presentation (R2, R3, R4), as we strive to make our findings accessible to a wider audience.

During rebuttal, we carefully addressed the main concerns raised by the reviewers. In particular, we clarified (1) the justification for using predicted results, (2) the rationale behind our agent construction and the simulation setting, and (3) the robustness of our conclusions.

In our revised manuscript, we have incorporated additional experiments and discussions, and relevant updates to further strengthen our work. Below, we summarize the core contributions of our study, the updates to our experiments, and the in-depth discussions included in our revision.

**Core Contributions of Our Work**

1. **Timely and critical topic: ​**LLMs  are  increasingly  used  to  refine  phrasing  or  even  generate manuscripts. On the other hand, some reviewers, despite explicit prohibitions, delegate their responsibilities to LLMs. In the current ICLR cycle, 21% of the review texts exhibited clear AI-generated characteristics [(source of information)](https://x.com/gneubig/status/1989681438577336401). An AI-generated submission received two reviewer scores of 8, suspected to be from LLM reviewers [(source of information)](https://x.com/micahgoldblum/status/1989088547777966512). LLM-generated submissions and reviewing are no longer isolated incidents but an emerging systemic issue.
2. **Novel Simulation Framework:** We introduce LLM-REVal, a multi-round simulation of the academic publication process, jointly modeling research and peer review to study systemic LLM reviewer behaviors in LLM-mediated scholarly ecosystems, examining the potential risks and unfairnesses.
3. **Identifying Misalignment:** We conduct human annotations and identify significant misalignments between LLM-based reviews and human judgments. These misalignments could severely undermine the fairness of peer review when LLMs are involved in the research process.
4. **Bias source:** We identify two primary biases in LLM reviewers: a linguistic feature bias favoring LLM-style writing and an aversion to critical statements (e.g., on topics like risk and fairness). These findings underscore the potential risks and fairness issues for human authors and academic research if LLMs are deployed in the peer review process without adequate caution.

**Updates of experimental results & in-depth discussions during Rebuttal**

We have corrected spelling and formatting errors throughout the text and in the figures. We would like to reiterate that all of the figures in our paper were manually created, with no AI involvement.

`Section 5.3`: Moved additional experiments on different models as LLM reviewers from the appendix into the main text.

`Section 7.2`: A more detailed analysis of Irreducible Rejection has been included.

`Appendix F.1`: Added related work as suggested.

`Appendix F.2`: Added experiments demonstrating that predicted results do not affect our conclusions and related discussion.

`Appendix F.3`: Discussed idea consistency in the simulation.

`Appendix F.4`: Provided further details regarding human papers.

`Appendix F.5`: Discussed the differences between our work and automated scientific discovery (ASD) as well as fabricated scientific discovery detection.

`Appendix F.6`: Discussed human-LLM correlation and human noise.

`Appendix F.7`: Discussed the construction of the agent.

`Appendix F.8`: Discussed the review scope.

We believe these additions and clarifications comprehensively address the reviewers' concerns and enhance the overall quality of our manuscript. All revisions are highlighted in `yellow-colored` text for ease of reference.

Thank you again for your time and for your service to the community. Please let us know if any additional clarification is needed.

Best regards,

Authors

---

### Meta-Review · Area_Chair_C5n4 · 2026-01-05

**Summary:**

This paper investigates the impact of using LLMs as both research and review agents by simulating a multi-round academic workflow in which papers are generated, reviewed, revised, and re-reviewed. Reviewers generally agree that the topic is timely and important, especially given growing interest in automated scientific discovery and AI-assisted peer review. Several reviewers appreciate the builder–reviewer loop design and the attempt to study feedback dynamics between AI-generated research and AI reviewers, which is relatively underexplored (Reviewer 9d5H and TMen). The empirical findings that LLM reviewers systematically favor LLM-written papers, reward stylistic conformity, and penalize human-authored papers that are self-critical or emphasize risks and fairness, are consistently observed and considered interesting and potentially impactful for discussions about fairness and trust in AI-based reviewing (Reviewer soLf, EYY5 and TMen).

**Reviewer Concerns:**

After carefully reading the rebuttal process, the following concerns are not addressed:
- The unfair and insufficiently controlled comparison between human-authored and LLM-generated papers: LLM papers rely on “predicted” or fabricated experimental results rather than executed experiments, while human papers report real results, introducing major confounders that make causal claims about reviewer bias difficult to justify (Reviewer 9d5H, soLf and EYY5).
- Multiple reviewers also note that the study reaffirms a phenomenon already known in the literature (LLMs preferring their own generations) without offering enough novelty or deeper causal analysis beyond re-demonstrating it in a simulated setting (Reviewer soLf). - Methodologically, the review process omits evaluation of data and code, limiting the reviewer agent’s ability to detect hallucinated or unsound research and weakening claims about research quality versus mere presentation effects (Reviewer 9d5H).
- Additional concerns include simplified or unrealistic research agents, unclear handling of disagreement between human and LLM reviewers (e.g., framing disagreement as “LLM bias” despite moderate correlation), over-sampling or dataset bias, and presentation issues such as an error-ridden architecture figure (Reviewer 9d5H, soLf, and EYY5).

**Reviewer Scores:**

- Reviewer 9d5H: This reviewer’s concerns are fundamental, focusing on missing evaluation of data and code, inability to distinguish real versus hallucinated research, and weaknesses in the overall simulation setup. These issues are structural and unlikely to be resolved through discussion, so a score increase is improbable.

- Reviewer soLf: Although generally positive about the topic, this reviewer emphasizes confounding factors between human and LLM-generated papers and limited novelty given prior findings. In discussion, convergence with similar critiques from other reviewers would likely reinforce caution rather than lead to a higher score.

- Reviewer EYY5: This reviewer finds the results interesting but highlights unfair comparisons and oversimplified research agents as key weaknesses. These concerns are echoed by others and would likely remain unresolved after discussion.

- Reviewer TMen: Strongly supportive of the paper’s novelty and framing.

---

### Decision · Program_Chairs · 2026-01-26

Reject